# Primal-Attention: Self-attention through Asymmetric Kernel SVD in Primal Representation

**Yingyi Chen** [*]
ESAT-STADIUS
KU Leuven, Belgium
`yingyi.chen@esat.kuleuven.be`

**Qinghua Tao** [*]
ESAT-STADIUS
KU Leuven, Belgium
`qinghua.tao@esat.kuleuven.be`

**Francesco Tonin**
ESAT-STADIUS
KU Leuven, Belgium
`francesco.tonin@esat.kuleuven.be`

**Johan A.K. Suykens**
ESAT-STADIUS
KU Leuven, Belgium
`johan.suykens@esat.kuleuven.be`

## Abstract

Recently, a new line of works has emerged to understand and improve self-attention in Transformers by treating it as a kernel machine. However, existing works apply the methods for symmetric kernels to the asymmetric self-attention, resulting in a nontrivial gap between the analytical understanding and numerical implementation. In this paper, we provide a new perspective to represent and optimize self-attention through asymmetric Kernel Singular Value Decomposition (KSVD), which is also motivated by the low-rank property of self-attention normally observed in deep layers. Through asymmetric KSVD, *i)* a primal-dual representation of self-attention is formulated, where the optimization objective is cast to maximize the projection variances in the attention outputs; *ii)* a novel attention mechanism, i.e., Primal-Attention, is proposed via the primal representation of KSVD, avoiding explicit computation of the kernel matrix in the dual; *iii)* with KKT conditions, we prove that the stationary solution to the KSVD optimization in Primal-Attention yields a zero-value objective. In this manner, KSVD optimization can be implemented by simply minimizing a regularization loss, so that low-rank property is promoted without extra decomposition. Numerical experiments show state-of-the-art performance of our Primal-Attention with improved efficiency. Moreover, we demonstrate that the deployed KSVD optimization regularizes Primal-Attention with a sharper singular value decay than that of the canonical self-attention, further verifying the great potential of our method. To the best of our knowledge, this is the first work that provides a *primal-dual representation* for the *asymmetric kernel* in self-attention and successfully applies it to *modelling* and *optimization*[2].

## 1   Introduction

Transformers [1] have become ubiquitous nowadays with state-of-the-art results in various tasks, such as natural language processing [2, 3, 4], computer vision [5, 6, 7, 8], reinforcement learning [9, 10, 11], etc. In the remarkable success of Transformers, the self-attention blocks play a key role, where the complicated dependencies between the individuals in data sequences can be depicted by using the established queries, keys, and values. Despite the prevailing advantages, theoretical understandings towards Transformers seem yet lagged behind its unprecedented empirical performance.

---

[*]Equal contribution.

[2]Our implementation is available at `https://github.com/yingyichen-cyy/PrimalAttention`

37th Conference on Neural Information Processing Systems (NeurIPS 2023).

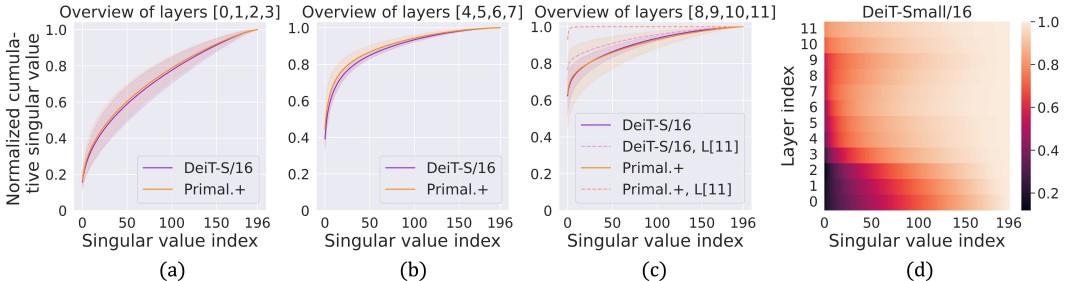

Figure 1: Spectrum analysis of the self-attention matrix on ImageNet-1K [23]. (a)-(c) plot the cumulative explained variance regarding the singular values of the attention matrix with mean and standard deviation of the chosen layers in pre-trained DeiT-Small/16 [7] and Primal.+DeiT-Small/16 (ours): the attention matrix attains sharper singular value decays in deeper layers, also shown in (d). Note that we also plot the cumulative explained variance curves of the self-attention matrix from the last layer, i.e., the 11-th layer denoted by "L[11]", of both models in (c). Our method shows an enhanced low-rank property of the attention matrix upon the baseline.

Recently, the kernel-based perspective has been proposed where the dot-product attention operation is shown as a kernel matrix [12]. This finding is quite encouraging by bridging kernels [13] with Transformers, as kernel methods have long been well studied with good interpretation ability. Following this spirit, different works have been proposed subsequently to improve self-attention, e.g., [14, 15, 16, 17, 18, 19]. However, in these works, the applied kernel techniques rely on Mercer kernels [20] requesting symmetry, which is inconsistent with the intrinsically asymmetric setups of self-attention. In [21], it analytically characterizes the attention by asymmetric kernels based on Reproducing Kernel Banach Spaces (RKBS) [22]. Nonetheless, neither the asymmetry property nor the related optimization is utilized for improvements. In [19], self-attention is derived with a primal-dual representation from the support vector regression, which still adopts the technique for Mercer kernels. Moreover, in the cast supervised task, the assumed ground-truth outputs of self-attention are practically non-existent, making it difficult to be applied in the optimization.

In this work, we provide a novel perspective to interpret self-attention with a primal-dual representation based on asymmetric Kernel Singular Value Decomposition (KSVD), which fills the gap of dismissing the asymmetry between theory and implementation. Specifically, in this unsupervised setup, we propose to remodel self-attention in the primal representation, namely, Primal-Attention, and to optimize it accordingly. Our method is driven by two major motivations. Firstly, we observe that attention matrices in Transformers can be low-rank, as shown in Figure 1(d), and this property becomes more significant towards deeper network layers. Secondly, the self-attention matrix is intrinsically an asymmetric kernel matrix [12, 21]. To this end, we propose KSVD for self-attention, which takes both low-rank and asymmetric properties into consideration. To the best of our knowledge, this is the first work that provides a primal-dual representation for the asymmetric self-attention and applies it to modelling and optimization. The contributions of this work are summarized as follows:

- We characterize self-attention by KSVD with asymmetric kernels. Different from existing works employing symmetric kernel-based methods, we take asymmetry into account so as to be more consistent with the real setups in self-attention. (Section 2)

- We derive a primal-dual representation for self-attention through KSVD, and propose a novel attention in the primal, named Primal-Attention, avoiding the expensive kernel computation in the dual. With KSVD, the values are interpreted as the projection weights that yield maximal variances of features, and the low-rank property can be pursued by confining the projection numbers. (Section 3, Section 4)

- We prove that the stationary solution to the derived KSVD leads to a zero-value objective of the unconstrained primal problem. Therefore, the optimization of KSVD in Primal-Attention can be efficiently implemented by minimizing a regularization term added to the loss, with no need of extra decomposition operations. (Section 4)

- In numerical experiments, Primal-Attention achieves state-of-the-art performance on various datasets together with efficiency advantages over the canonical self-attention. Moreover, we demonstrate that our deployed optimization from KSVD can regularize the attention with a

sharper singular value decay, hence promoting learning more low-rank features, which is shown in Figure 1. (Section 5)

## 2 Problem Statement: Self-attention with Asymmetric Kernel

**Self-attention** Let $\{\boldsymbol{x}_i \in \mathbb{R}^d\}_{i=1}^N$ be the input data sequence. In self-attention [1], the queries, keys and values output the linear projections of the input sequence, such that

$$q(\boldsymbol{x}_i) = W_q \boldsymbol{x}_i, \quad k(\boldsymbol{x}_i) = W_k \boldsymbol{x}_i, \quad v(\boldsymbol{x}_i) = W_v \boldsymbol{x}_i, \tag{1}$$

where $W_q \in \mathbb{R}^{d_q \times d}$, $W_k \in \mathbb{R}^{d_k \times d}$, and $W_v \in \mathbb{R}^{d_v \times d}$, commonly with the setup $d_q = d_k$. The attention scores are then given by $a(\boldsymbol{x}_i, \boldsymbol{x}_j) = \langle q(\boldsymbol{x}_i), k(\boldsymbol{x}_j) \rangle / \sqrt{d_k} = \langle W_q \boldsymbol{x}_i, W_k \boldsymbol{x}_j \rangle / \sqrt{d_k}$. In the canonical self-attention, the "softmax" activation is then applied to bring non-linearity and positives, yielding the attention weights:

$$\kappa(\boldsymbol{x}_i, \boldsymbol{x}_j) = \text{softmax}\left(\langle W_q \boldsymbol{x}_i, W_k \boldsymbol{x}_j \rangle / \sqrt{d_k}\right), \quad i, j = 1, \ldots, N. \tag{2}$$

Similar to [12], the attention matrix, i.e., $K := [\kappa(\boldsymbol{x}_i, \boldsymbol{x}_j)] \in \mathbb{R}^{N \times N}$, can be interpreted as a kernel matrix with entries $\kappa(\boldsymbol{x}_i, \boldsymbol{x}_j)$, where $\kappa(\cdot, \cdot) \colon \mathbb{R}^d \times \mathbb{R}^d \to \mathbb{R}$ serves as the kernel function. Notice that in general, $\langle W_q \boldsymbol{x}_i, W_k \boldsymbol{x}_j \rangle \neq \langle W_q \boldsymbol{x}_j, W_k \boldsymbol{x}_i \rangle$, leading to an asymmetric kernel where $K_{ij} \neq K_{ji}$.

Then, the attention output $\boldsymbol{o}_i \in \mathbb{R}^{d_v}$ in each head is attained as:

$$\boldsymbol{o}_i = \sum_{j=1}^N v(\boldsymbol{x}_j)\kappa(\boldsymbol{x}_i, \boldsymbol{x}_j) = \sum_{j=1}^N v(\boldsymbol{x}_j)K_{ij}, \quad i = 1, \ldots, N. \tag{3}$$

In Transformers, multiple heads are commonly applied through the concatenation of all heads [1].

**Asymmetric Attention Matrix** In kernel methods, rigorous works have been presented with Mercer kernels that are symmetric and positive semi-definite [20] through the kernel trick from Reproducing Kernel Hilbert Spaces (RKHS) [13]. On the other hand in Transformer [1], the attention kernel matrix is asymmetric as shown in (2). Existing works leverage the kernel interpretation for improving self-attention [14, 17, 18, 19], however, their deployed kernel-based techniques all rely on Mercer kernels, which is inconsistent with the asymmetric nature. Instead, asymmetry is allowed in kernel tricks from Reproducing Kernel Banach Spaces (RKBS) [22] as in the following Definition 2.1.

**Definition 2.1** (Definition 2 [21]; Theorem 2.1 [24];[25])**.** *For asymmetric kernels, the kernel trick from RKBS with the kernel function $\kappa(\cdot, \cdot) \colon \mathcal{X} \times \mathcal{Z} \to \mathbb{R}$ can be defined by the inner product of two real measurable feature maps from Banach spaces $\mathcal{B}_\mathcal{X}, \mathcal{B}_\mathcal{Z}$ on $\mathcal{X}, \mathcal{Z}$, respectively:*

$$\kappa(\boldsymbol{x}, \boldsymbol{z}) = \langle \phi_x(\boldsymbol{x}), \phi_z(\boldsymbol{z}) \rangle, \forall \boldsymbol{x} \in \mathcal{X}, \phi_x \in \mathcal{B}_\mathcal{X}, \boldsymbol{z} \in \mathcal{Z}, \phi_z \in \mathcal{B}_\mathcal{Z}. \tag{4}$$

Based on Definition 2.1, the kernel matrix in self-attention can be characterized by the kernel trick from RKBS [21], providing an analytical tool from the aspect of kernel representer theorem.

**SVD and Shifted Eigenvalue Problem** SVD factorizes a given $r$-rank matrix $A \in \mathbb{R}^{N \times M}$ by two sets of orthonormal eigenbases: $A = U\Sigma V^\top$ with $\Sigma = \text{diag}\{\sigma_1, \ldots, \sigma_r\}$ of positive singular values and the columns of $U \in \mathbb{R}^{N \times r}$ and $V \in \mathbb{R}^{M \times r}$ as the *left* and *right singular vectors*, respectively [26]. $U, V$ reflect the subspace projections in relation to the columns and rows, as shown in (5), and contain different information residing in $A$ due to the asymmetry. When $A$ is squared and symmetric, SVD boils down to the eigendecomposition with $U = V$. In [27], a novel variational principle is proposed for SVD with Least Squares Support Vector Machines (LSSVM) [28], where the dual problem leads to a shifted eigenvalue problem in accordance to the decomposition theorem from Lanczos [29] regarding SVD, i.e., Theorem 2.2. This theorem is also of special importance in our work to the kernel extension of SVD in self-attention under the framework of LSSVM.

**Theorem 2.2** (Lanczos [29])**.** *Any non-zero matrix $A \in \mathbb{R}^{N \times M}$ can be written as $A = \tilde{U}\tilde{\Sigma}\tilde{V}^\top$, where the matrices $\tilde{U}, \tilde{\Sigma}, \tilde{V}$ are defined by the shifted eigenvalue problem:*

$$\begin{aligned} A\tilde{V} &= \tilde{U}\tilde{\Sigma}, \\ A^\top \tilde{U} &= \tilde{V}\tilde{\Sigma}, \end{aligned} \tag{5}$$

*where $\tilde{U} \in \mathbb{R}^{N \times r}$ and $\tilde{V} \in \mathbb{R}^{M \times r}$ satisfy $\tilde{U}^\top \tilde{U} = I_r$ and $\tilde{V}^\top \tilde{V} = I_r$, and $\tilde{\Sigma} \in \mathbb{R}^{r \times r}$ is a diagonal matrix with positive numbers.*

# 3 Primal-dual Representation of Self-attention based on Kernel SVD

In this section, we apply the kernel trick from RKBS to the asymmetric attention kernel, and derive self-attention with a primal-dual representation based on Kernel SVD (KSVD). Under this learning scheme, a new self-attention mechanism is proposed by remodelling the attention output in the primal representation, without explicit computation of the kernel matrix in the dual representation. With the stationarity conditions, we flexibly implement the optimization of KSVD through an additional loss term, which can regularize the model to improved low-rank properties without extra decomposition.

**KSVD optimization problem in Primal and Dual**  By Definition 2.1 of RKBS, the kernel function in the dual for the asymmetric attention kernel $K$ in self-attention can be formulated by $K_{ij} = \kappa(\boldsymbol{x}_i, \boldsymbol{x}_j) := \langle \phi_q(\boldsymbol{x}_i), \phi_k(\boldsymbol{x}_j) \rangle$, with two feature maps $\phi_q$, $\phi_k$ related to queries and keys. Recall the self-attention output in (3), the values $\{v(\boldsymbol{x}_j)\}_{j=1}^N$ resemble the dual variables projecting the kernel matrix in the dual representation of kernel methods, whereas the kernel involved is asymmetric. In this regard, the nonlinear version of SVD under the framework of LSSVM [27] well fits the self-attention setup, hence is set as the basis for the following work with asymmetric kernels built upon RKBS. Differently, we extend the matrix SVD setups in [27] to the case of asymmetric attention matrix with two input data sources as queries and keys. Moreover, we consider that in self-attention, values are input data-dependent, and thus generalize [27] with data-dependent projection weights.

Given the sequence $\{\boldsymbol{x}_i \in \mathbb{R}^d\}_{i=1}^N$, we start from the primal optimization with KSVD:

$$
\begin{aligned}
\max_{W_e, W_r, \boldsymbol{e}_i, \boldsymbol{r}_j} \quad & J = \frac{1}{2} \sum_{i=1}^N \boldsymbol{e}_i^\top \Lambda \boldsymbol{e}_i + \frac{1}{2} \sum_{j=1}^N \boldsymbol{r}_j^\top \Lambda \boldsymbol{r}_j - \mathrm{Tr}\left(W_e^\top W_r\right) \\
\text{s.t.} \quad & \boldsymbol{e}_i = (f(X)^\top W_e)^\top \phi_q(\boldsymbol{x}_i), \ i = 1, \dots, N, \\
& \boldsymbol{r}_j = (f(X)^\top W_r)^\top \phi_k(\boldsymbol{x}_j), \ j = 1, \dots, N,
\end{aligned}
\tag{6}
$$

where we have the data-dependent projection weights $f(X)^\top W_e =: W_{e|X} \in \mathbb{R}^{p \times s}$, $f(X)^\top W_r =: W_{r|X} \in \mathbb{R}^{p \times s}$ relying on parameters $W_e, W_r \in \mathbb{R}^{N \times s}$, the feature maps $\phi_q(\cdot), \phi_k(\cdot) \colon \mathbb{R}^d \to \mathbb{R}^p$, the projection scores $\boldsymbol{e}_i, \boldsymbol{r}_j \in \mathbb{R}^s$, and the regularization coefficient $\Lambda \in \mathbb{R}^{s \times s}$ which is a positive diagonal matrix. The objective $J$ in the primal optimization maximizes the projection variances of $W_{e|X}^\top \phi_q(\boldsymbol{x}_i), W_{r|X}^\top \phi_k(\boldsymbol{x}_j)$ regarding queries and keys, and also involves a regularization term coupling the projections. The corresponding solution in the dual is characterized by the right and left singular vectors, which captures the directions with maximal projection variances, w.r.t. rows (queries) and columns (keys) of the attention kernel matrix. Thus, with the formulated primal optimization in (6), the learning in self-attention is interpreted as a SVD problem on the attention matrix.

For clarity, we elaborate our primal optimization problem in (6) as follows. *i)* The projection weights are data-dependent, where $f(X) =: F_X \in \mathbb{R}^{N \times p}$ denotes a transformation matrix containing the information of the sequence data $X := [\boldsymbol{x}_1, \dots, \boldsymbol{x}_N]^\top \in \mathbb{R}^{N \times d}$. Notably, $F_X$ is a constant matrix once given $X$, and we let it linearly depend on $X$ in experiments. Moreover, when $F_X$ is chosen as the identity matrix, it reconciles to the common setups in kernel methods in primal. *ii)* The feature maps related to queries and keys, respectively, are defined as $\phi_q(\boldsymbol{x}_i) := g_q(q(\boldsymbol{x}_i)), \phi_k(\boldsymbol{x}_i) := g_k(k(\boldsymbol{x}_i))$, where $g_q(\cdot) : \mathbb{R}^{d_q} \to \mathbb{R}^p$ and $g_k(\cdot) : \mathbb{R}^{d_k} \to \mathbb{R}^p$ denote the mappings composited on the linear projections $q(\cdot)$ and $k(\cdot)$ in (1) of queries and keys. Note that we leave the choice of $g_q$ and $g_k$ later explained in Remark 4.1. *iii)* By projecting $\phi_q(\boldsymbol{x}_i), \phi_k(\boldsymbol{x}_j) \in \mathbb{R}^p$ with weights $W_{e|X}, W_{r|X} \in \mathbb{R}^{p \times s}$, we obtain the projection scores $\boldsymbol{e}_i, \boldsymbol{r}_j \in \mathbb{R}^s$ along the $s$ directions, usually $s < p$, which corresponds to the number of singular values of the induced kernel matrix in the dual optimization (7).

**Remark 3.1** (Variance maximization objective). *In the formulated KSVD problem, the objective in the primal optimization* (6) *jointly maximizes the variances of the two projections $\boldsymbol{e}_i, \boldsymbol{r}_j$ in the feature spaces determined by $\phi_q$, $\phi_k$ along the directions $W_{e|X}, W_{r|X}$. Within this context, $\boldsymbol{e}_i, \boldsymbol{r}_j$ learn to capture maximal information mutually residing in $\phi_q$ and $\phi_k$ regarding the queries and keys.*

With Lagrangian duality and KKT conditions, we prove that the dual optimization problem to (6) leads to a shifted eigenvalue problem corresponding to the SVD on the asymmetric attention kernel $K$, given by Theorem 3.2. The proof is provided in the Supplementary Material.

**Theorem 3.2** (Dual optimization problem of KSVD in self-attention). *With Lagrangian duality and the KKT conditions, the dual optimization problem of* (6) *leads to the shifted eigenvalue problem:*

$$
\begin{aligned}
K H_r &= H_e \Sigma, \\
K^\top H_e &= H_r \Sigma,
\end{aligned}
\tag{7}
$$

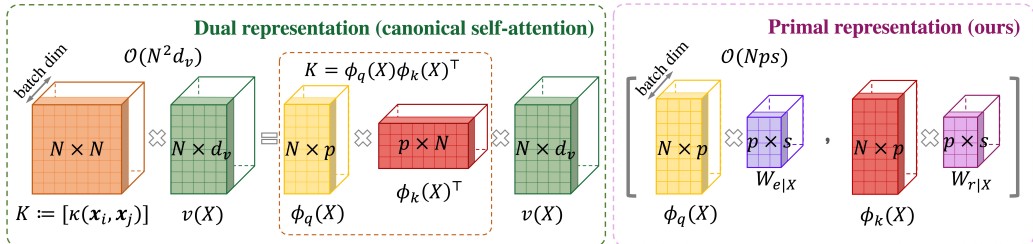

Figure 2: An illustration of Primal-Attention and canonical self-attention. Left: in *canonical self-attention*, the asymmetric attention matrix $K$ can be induced by two feature maps $\phi_q$, $\phi_k$ through kernel trick from RKBS. The values $v(X)$ serve as the dual variables projecting the kernel matrix $K$ to the attention output. The time and space complexity are $\mathcal{O}(N^2 d_v)$ and $\mathcal{O}(N^2 + Nd_v)$. Right: in our *Primal-Attention*, we use the two feature maps $\phi_q$, $\phi_k$ in the primal to present the attention outputs, which are projected through the primal variables $W_{e|X}$, $W_{r|X}$. The time and space complexity are $\mathcal{O}(Nps)$ and $\mathcal{O}(2Np + 2ps)$. To align with the output size in the canonical self-attention, we add a linear map mapping from $2s$ to $d_v$ with negligible memory increase after Primal-Attention's output.

where $\Sigma \in \mathbb{R}^{s \times s}$ is a positive diagonal matrix, and $H_e = [\boldsymbol{h}_{e_1}, \ldots, \boldsymbol{h}_{e_N}]^\top \in \mathbb{R}^{N \times s}$, $H_r = [\boldsymbol{h}_{r_1}, \ldots, \boldsymbol{h}_{r_N}]^\top \in \mathbb{R}^{N \times s}$ *are the dual variables serving as the left and right singular vectors, respectively. The kernel trick to the asymmetric kernel matrix $K$ is interpreted as $K_{ij} = \langle f(X)g_q(q(\boldsymbol{x}_i)), f(X)g_k(k(\boldsymbol{x}_j)) \rangle =: \langle \phi'_q(\boldsymbol{x}_i), \phi'_k(\boldsymbol{x}_j) \rangle$.*

As shown in Theorem 3.2, the solutions collect non-zero $\Lambda$ in (6) such that $\Sigma = \Lambda^{-1}$. Based on Lanczos' decomposition in Theorem 2.2, we can then associate $\Sigma$ with the non-zero singular values of the attention kernel $K$ in (7), and $H_e, H_r$ with the left and right singular vectors of $K$, such that $K = H_e \Sigma H_r^\top$. Therefore, formulas (6) and (7) provide the optimization problems of performing KSVD with the attention kernel matrix $K$ in the primal and in the dual, respectively.

**Self-attention as KSVD dual representation**  Firstly, we provide the primal-dual model representation of the derived KSVD problem. Secondly, we show that the dual representation of the model corresponds to the canonical self-attention. The derivation details involving the KKT conditions are provided in the Supplementary Material.

**Remark 3.3** (Primal-dual representations of KSVD in self-attention). *In the KSVD formulations for the asymmetric kernel matrix in self-attention, with KKT conditions, the projection scores can be either represented in the primal using explicit feature maps or in the dual using kernel functions:*

$$\textit{Primal:} \begin{cases} e(\boldsymbol{x}) = W_{e|X}^\top \phi_q(\boldsymbol{x}) \\ r(\boldsymbol{x}) = W_{r|X}^\top \phi_k(\boldsymbol{x}) \end{cases}, \quad \textit{Dual:} \begin{cases} e(\boldsymbol{x}) = \sum_{j=1}^N \boldsymbol{h}_{r_j} \kappa(\boldsymbol{x}, \boldsymbol{x}_j) \\ r(\boldsymbol{x}) = \sum_{i=1}^N \boldsymbol{h}_{e_i} \kappa(\boldsymbol{x}_i, \boldsymbol{x}). \end{cases} \quad (8)$$

**Remark 3.4** (Correspondence of KSVD and canonical self-attention output). *Recall the output $\boldsymbol{o}_i$ of canonical self-attention (3), it corresponds to the dual representation of the projection score $e(\boldsymbol{x})$ in (8), i.e., $\boldsymbol{o}_i \triangleq e(\boldsymbol{x}_i)$. When the values $\{v(\boldsymbol{x}_j)\}_{j=1}^N$ in canonical self-attention are chosen as the dual variables $\{\boldsymbol{h}_{r_j}\}_{j=1}^N$, i.e., $\boldsymbol{h}_{r_j} := v(\boldsymbol{x}_j)$, $j = 1, \ldots, N$, the values $\{v(\boldsymbol{x}_j)\}_{j=1}^N$ play the role of the right singular vectors of $K$.*

From the perspective in Remark 3.4, the optimization goal in self-attention is interpreted to jointly capture the maximal variances of $\boldsymbol{e}_i$, $\boldsymbol{r}_j$ as in (6), where the projection scores can be denoted as $\boldsymbol{e}_i := e(\boldsymbol{x}_i)$, $\boldsymbol{r}_j := r(\boldsymbol{x}_j)$ through the representations (8). However, the canonical self-attention only outputs the $\boldsymbol{e}_i$-score ($\boldsymbol{o}_i \triangleq \boldsymbol{e}_i$). In this sense, the output of the canonical self-attention only considers the projection scores involving the right singular vectors of the asymmetric attention kernel $K$.

## 4  Primal-Attention

**Modelling**  It is quite remarkable that the attention output can be equivalently represented without the kernel expressions, avoiding the heavy computation of the kernel matrix. Within the context of KSVD, we further observe that there exists another set of projections $\boldsymbol{r}_j$ regarding the left singular vectors in $\boldsymbol{h}_{e_i}$ as in (8), providing extra information residing in the asymmetric kernel matrix $K$.

We derive a novel attention mechanism by leveraging the primal representation of KSVD, namely, Primal-Attention, where two explicit feature maps $\phi_q, \phi_k$ are adopted. To fully exploit the asymmetry in the kernel matrix of self-attention, Primal-Attention concatenates the two sets of projections using both left and right singular vectors, and thus formulates the attention outputs as follows:

$$\boldsymbol{o}_i := [\boldsymbol{e}_i; \boldsymbol{r}_i] = \left[ W_{e|X}^\top \phi_q(\boldsymbol{x}_i); W_{r|X}^\top \phi_k(\boldsymbol{x}_i) \right] = \left[ W_e^\top f(X) g_q(q(\boldsymbol{x}_i)); W_r^\top f(X) g_k(k(\boldsymbol{x}_i)) \right]. \quad (9)$$

In Primal-Attention, the projection weights $W_{e|X}, W_{r|X}$ in the primal play the role as the counterparts of the values in the dual. Given $f(X) =: F_X$ an identity matrix, our KSVD problem in (6) boils down to the data-independent projection weight case as in [27], which can thereby be regarded as a special case of our derived KSVD in Primal-Attention. In this case, the kernel trick in the asymmetric attention kernel becomes $K_{ij} = \langle g_q(q(\boldsymbol{x}_i)), g_k(k(\boldsymbol{x}_j)) \rangle = \langle \phi_q(\boldsymbol{x}_i), \phi_k(\boldsymbol{x}_j) \rangle, i, j = 1, \ldots, N$.

**Remark 4.1** (Choices of $\phi_q, \phi_k$ for non-linearity). *The canonical self-attention adopts softmax for introducing non-linearity to the attention score matrix; within our setups and kernel trick, it can be viewed as: $\kappa(\boldsymbol{x}_i, \boldsymbol{x}_j) = \hat{D}^{-1} \langle \phi_q(\boldsymbol{x}_i), \phi_k(\boldsymbol{x}_i) \rangle$ where $\phi_q(\boldsymbol{x}) := g(q(\boldsymbol{x}))$, $\phi_k(\boldsymbol{x}) := g(k(\boldsymbol{x}))$ with $g(\boldsymbol{z}) := \exp(-\|\boldsymbol{z}\|^2/2)(\exp(\boldsymbol{w}_1^\top \boldsymbol{z}), \ldots, \exp(\boldsymbol{w}_p^\top \boldsymbol{z}))$, $\boldsymbol{w}_i \sim \mathcal{N}(0, I_{d_q})$ and $d_q = d_k$ [14], and $\hat{D} := \mathrm{diag}(\phi_q(X)(\phi_k(X)^\top \mathbf{1}_N))$. The projection scores then correspond to $e(\boldsymbol{x}) = \hat{D}^{-1/2} W_{e|X}^\top \phi_q(\boldsymbol{x})$ and $r(\boldsymbol{x}) = \hat{D}^{-1/2} W_{r|X}^\top \phi_k(\boldsymbol{x})$. In this case, two exponential feature maps need to be constructed and the normalization factor $\hat{D}$ to all samples needs to be computed. In this paper, we consider feature maps related to the Cosine similarity kernel on queries and keys, such that $\phi_q(\boldsymbol{x}) = g_q(q(\boldsymbol{x})) := q(\boldsymbol{x})/\|q(\boldsymbol{x})\|_2$ and $\phi_k(\boldsymbol{x}) = g_k(k(\boldsymbol{x})) := k(\boldsymbol{x})/\|k(\boldsymbol{x})\|_2$ in all experiments. It is easy to implement and able to bring both non-linearity and normalization to the feature maps.*

**Optimization** The KSVD problem can either be optimized in the primal as a constrained optimization problem, or in the dual as a shifted eigenvalue problem (SVD on the kernel matrix $K$). In Primal-Attention, we perform the optimization in the primal. In the following Lemma 4.2, we prove the zero-value property of the primal optimization objective $J$ in (6) when it is evaluated at the stationary solution in (7). The proof is provided in the Supplementary Material.

**Lemma 4.2** (A zero-value objective with stationary solutions). *The solutions of $H_e, H_r, \Sigma$ to the shifted eigenvalue problem in the dual optimization (7) lead to the zero-value objective $J$ in the primal optimization (6).*

With the property in Lemma 4.2, rather than solving SVD problem on the kernel matrix $K$ in the dual, the optimization of Primal-Attention is realized by minimizing the primal objective to zero:

$$\min \ L + \eta \sum_l J_l^2, \quad (10)$$

where $L$ is the task-oriented loss, e.g., the cross-entropy loss for classification tasks, the summation term over $l$ denotes the additive objectives $J_l$ of all attention blocks using the proposed Primal-Attention, where $J_l$ is implemented as the mean over all heads, and $\eta > 0$ is the regularization coefficient. Specifically, for each head in the self-attention using Primal-Attention, we have

$$\begin{aligned} J(W_e, W_r, \Lambda) &= \tfrac{1}{2} \sum_{i=1}^N \boldsymbol{e}_i^\top \Lambda \boldsymbol{e}_i + \tfrac{1}{2} \sum_{j=1}^N \boldsymbol{r}_j^\top \Lambda \boldsymbol{r}_j - \mathrm{Tr}\left( W_e^\top W_r \right) \\ &= \tfrac{1}{2} \sum_{i=1}^N \|(W_{e|X} \Lambda^{\frac{1}{2}})^\top \phi_q(\boldsymbol{x}_i)\|_2^2 + \tfrac{1}{2} \sum_{j=1}^N \|(W_{r|X} \Lambda^{\frac{1}{2}})^\top \phi_k(\boldsymbol{x}_j)\|_2^2 - \mathrm{Tr}\left( W_e^\top W_r \right), \end{aligned} \quad (11)$$

where $\Lambda$ is automatically determined by the optimization. KSVD optimization of Primal-Attention is easy to implement by adding a regularization loss. Hence, Primal-Attention not only represents self-attention with KSVD formulation in the primal, but utilizes the optimization of KSVD by regularizing the attention, satisfying the stationarity conditions of KSVD when reaching a zero-value objective.

## 5  Numerical Experiments

In this section, we verify the effectiveness of our Primal-Attention applied in Transformers on five well-established benchmarks: time series, long sequence modelling, reinforcement learning, image classification and language modelling. Notably, we consider two types of Transformers applied with our Primal-Attention, i.e., PrimalFormer (Primal.) and Primal.+. *i)* In PrimalFormer, Primal-Attention (9) is applied to all attention layers, regularized with the KSVD loss (11). This setup is

Table 1: Test accuracy (%) on the UEA time series classification archive benchmark [31] with comparisons to canonical Transformer (Trans.), Linear Transformer (Linear.), Reformer (Re.), Longformer (Long.), Performer (Per.), cosFormer (Cos.), Flowformer (Flow.), YOSO-E and SOFT.

| Dataset / Model | Trans. | Linear. | Re. | Long. | Per. | YOSO-E | Cos. | SOFT | Flow. | Ours | |
|---|---|---|---|---|---|---|---|---|---|---|---|
| | [1] | [33] | [34] | [35] | [14] | [36] | [37] | [38] | [11] | Primal. | Primal.+Trans. |
| EthanolConcentration | 32.7 | 31.9 | 31.9 | 32.3 | 31.2 | 31.2 | 32.3 | 33.5 | 33.8 | 33.1 | 35.4 |
| FaceDetection | 67.3 | 67.0 | 68.6 | 62.6 | 67.0 | 67.3 | 64.8 | 67.1 | 67.6 | 67.1 | 63.8 |
| HandWriting | 32.0 | 34.7 | 27.4 | 39.6 | 32.1 | 30.9 | 28.9 | 34.7 | 33.8 | 29.6 | 28.7 |
| HeartBeat | 76.1 | 76.6 | 77.1 | 78.0 | 75.6 | 76.5 | 77.1 | 75.6 | 77.6 | 76.1 | 77.1 |
| JapaneseVowels | 98.7 | 99.2 | 97.8 | 98.9 | 98.1 | 98.6 | 98.3 | 99.2 | 98.9 | 98.4 | 98.9 |
| PEMS-SF | 82.1 | 82.1 | 82.7 | 83.8 | 80.9 | 85.2 | 83.2 | 80.9 | 83.8 | 89.6 | 90.2 |
| SelfRegulationSCP1 | 92.2 | 92.5 | 90.4 | 90.1 | 91.5 | 91.1 | 91.1 | 91.8 | 92.5 | 92.5 | 92.5 |
| SelfRegulationSCP2 | 53.9 | 56.7 | 56.7 | 55.6 | 56.7 | 53.9 | 55.0 | 55.6 | 56.1 | 57.2 | 56.1 |
| SpokenArabicDigits | 98.4 | 98.0 | 97.0 | 94.4 | 98.4 | 98.9 | 98.4 | 98.8 | 98.8 | 100 | 100 |
| UWaveGestureLibrary | 85.6 | 85.0 | 85.6 | 87.5 | 85.3 | 88.4 | 85.6 | 85.0 | 86.6 | 86.3 | 88.4 |
| Average Accuracy | 71.9 | 72.4 | 71.5 | 72.0 | 71.9 | 72.2 | 71.5 | 72.2 | 73.0 | 73.0 | **73.1** |

Table 2: Efficiency comparisons on running time and memory consumption on the UEA benchmark [31] where running time (s/Epoch), the peak training memory usage (GB) are given. We report how much faster and how less memory each model uses than Transformer.

| Dataset / seq. length | Ethanol. 1751 | Face. 62 | Hand. 152 | Heart. 405 | Jap. 26 | PEMS-SF 144 | SCP1 896 | SCP2 1152 | Arabic. 83 | UWave 315 | Avg. Time (s/Epoch) | Memory (GB) |
|---|---|---|---|---|---|---|---|---|---|---|---|---|
| Trans. [1] | 4.3 (1×) | 10.7 (1×) | 0.3 (1×) | 0.7 (1×) | 0.5 (1×) | 0.6 (1×) | 1.8 (1×) | 1.8 (1×) | 3.7 (1×) | 0.3 (1×) | 2.5 (1×) | 10.9 (1×) |
| Flow. [11] | 2.4 (1.8×) | 9.8 (1.1×) | 0.3 (1.0×) | 0.7 (1.0×) | 0.6 (0.8×) | 0.7 (0.9×) | 1.4 (1.3×) | 1.3 (1.4×) | 4.4 (0.8×) | 0.3 (1.0×) | 2.2 (1.1×) | 2.8 (3.9×) |
| **Primal.+Trans.** | 3.3 (1.3×) | **5.7 (1.9×)** | 0.3 (1.0×) | 0.7 (1.0×) | 0.5 (1.0×) | 0.7 (0.9×) | 1.6 (1.1×) | 1.6 (1.1×) | 4.0 (0.9×) | 0.3 (1.0×) | **1.9 (1.3×)** | 6.5 (1.7×) |
| **Primal.** | **2.3 (1.9×)** | 6.9 (1.6×) | 0.4 (0.8×) | **0.4 (1.8×)** | 0.5 (1.0×) | 0.8 (0.8×) | **1.4 (1.3×)** | **1.3 (1.4×)** | 4.5 (0.8×) | 0.4 (0.8×) | **1.9 (1.3×)** | **2.7 (4.0×)** |

preferred when data shows relatively strong low-rank structure or the model redundancy is quite substantial. *ii)* Primal.+ refers to the baselines from the Transformer family with the last layer replaced by our Primal-Attention, which is in favour in large-scale data and complicated tasks where less information compression is desired in the learning, especially in shallower layers. To specify the Transformer backbone, we denote our method by "Primal.+Backbone" herein. Primal.+ serves as a flexible variant combined with different Transformer backbones, and with KSVD optimization applied through an implicit low-rank regularization loss, advocating to learn more informative features, as in Figure 1. The two main hyper-parameters of our method are the coefficient $\eta$ in (10) and the number of projection directions $s$ of KSVD in (6). In data-dependent cases with $f(X)$, we take a subset of $X$ by uniformly sampling $n = \min\{s*10, N\}$ points from $X$ for efficiency aspects, as the main patterns of a matrix can be retained with random linear projections shown by the Johnson–Lindenstrauss lemma [30]. Details are given in the Supplementary Material.

**UEA Time Series Classification**  UEA Time Series Classification Archive [31] is the benchmark for the evaluation on temporal sequences. Following [11], we select 10 multivariate datasets with pre-processed data according to [32], and employ 2-layer Transformer as backbone with the hidden dimension 512 on 8 heads and the embedding dimension 64 for self-attention. The hyper-parameter search of our method is with $\eta \in \{0.1, 0.2, 0.5\}$, $s \in \{20, 30, 40\}$. Experiments are run on one NVIDIA Tesla P100 SXM2 16GB GPU.

Table 1 reports the test accuracy of the compared recent methods, where the best result is in bold. Both our PrimalFormer and Primal.+ achieve comparable and better performance than the state-of-the-art results provided by Flowformer [11]. Notably, Primal.+ yields the best accuracy with 1.2% overall improvement upon the canonical Transformer [1], i.e., by replacing the softmax-based self-attention with our Primal-Attention in the last layer. This shows the promising potential of Primal-Attention in enhancing temporal modelling capacity upon the canonical softmax self-attention. It is worth mentioning that our PrimalFormer applying the KSVD optimization with low-rank regularization to all layers also obtains good performance. This could be due to the fact that these datasets are relatively simple where model redundancy can exist, so that appropriately imposing the low-rank regularization in KSVD does not harm the model expressiveness for the data. We also compare the running time and memory consumption with the canonical Transformer and Flowformer, which is proposed recently with state-of-the-art performances and efficiency. Both our Primal. and Primal.+Trans. consistently lead to improved efficiency than the canonical Transformer. In Primal., all attention layers are implemented with the proposed Primal-Attention, while Primal.+Trans. applies the Primal-Attention

Table 3: Test accuracy (%) on the LRA benchmark [39] with comparisons to canonical Transformer (Trans.), Reformer (Re.), Performer (Per.), Linformer (Lin.), Nyströmformer (Nyström.), Longformer (Long.) and YOSO-E.

| Dataset (seq. length) | Trans. | Re. | Per. | Lin. | Nyström. | Long. | YOSO-E | Ours | |
|---|---|---|---|---|---|---|---|---|---|
| | [1] | [34] | [14] | [46] | [45] | [35] | [36] | Primal. | Primal.+Trans. |
| ListOps (2K) | 37.1 | 19.1 | 18.8 | 37.3 | 37.2 | 37.2 | 37.3 | 37.3 | 37.3 |
| Text (4K) | 65.0 | 64.9 | 63.8 | 55.9 | 65.5 | 64.6 | 64.7 | 61.2 | 65.4 |
| Retrieval (4K) | 79.4 | 78.6 | 78.6 | 79.4 | 79.6 | 81.0 | 81.2 | 77.8 | 81.0 |
| Image (1K) | 38.2 | 43.3 | 37.1 | 37.8 | 41.6 | 39.1 | 39.8 | 43.0 | 43.9 |
| Pathfinder (1K) | 74.2 | 69.4 | 69.9 | 67.6 | 70.9 | 73.0 | 72.9 | 68.3 | 74.3 |
| Average Accuracy | 58.8 | 55.1 | 53.6 | 55.6 | 59.0 | 59.0 | 59.2 | 57.5 | **60.4** |

Table 4: Efficiency comparisons on running time and memory consumption on LRA [39].

| Model | Time (s/1K-steps) | | | | | Memory (GB) | | | | |
|---|---|---|---|---|---|---|---|---|---|---|
| | ListOps | Text | Retrieval | Image | Pathfinder | ListOps | Text | Retrieval | Image | Pathfinder |
| Trans. [1] | 194.5 (1×) | 694.8 (1×) | 1333.7 (1×) | 334.5 (1×) | 405.5 (1×) | 5.50 (1×) | 21.24 (1×) | 18.72 (1×) | 5.88 (1×) | 5.88 (1×) |
| Nyström. [45] | 68.4 (2.8×) | 120.9 (5.7×) | 235.5 (5.7×) | 179.5 (1.9×) | 221.2 (1.8×) | 0.89 (6.2×) | 1.69 (12.6×) | 3.29 (5.7×) | 1.93 (3.0×) | 1.93 (3.0×) |
| Lin. [46] | 63.4 (3.1×) | 116.5 (6.0×) | 226.2 (5.9×) | 158.5 (2.1×) | 204.0 (2.0×) | 1.73 (3.2×) | 3.45 (6.2×) | 6.33 (3.0×) | 3.45 (1.7×) | 3.45 (1.7×) |
| Per. [14] | 83.8 (2.3×) | 157.5 (4.4×) | 320.6 (4.2×) | 211.4 (1.6×) | 278.1 (1.5×) | 1.67 (3.3×) | 3.34 (6.4×) | 6.28 (3.0×) | 3.34 (1.8×) | 3.34 (1.8×) |
| Re. [34] | 87.0 (2.2×) | 168.5 (4.1×) | 339.9 (3.9×) | 223.7 (1.5×) | 286.7 (1.4×) | 1.64 (3.3×) | 3.29 (6.5×) | 6.09 (3.1×) | 3.29 (1.8×) | 3.29 (1.8×) |
| **Primal.+Trans.** | 113.4 (1.7×) | 367.6 (1.9×) | 546.5 (2.4×) | 212.1 (1.6×) | 263.2 (1.5×) | 5.24 (1.1×) | 20.7 (1.0×) | 18.59 (1.0×) | 5.35 (1.1×) | 5.35 (1.1×) |
| **Primal.** | 56.5 (3.4×) | 93.6 (7.4×) | 185.3 (7.2×) | 142.9 (2.3×) | 180.0 (2.3×) | 0.69 (7.9×) | 1.37 (15.5×) | 2.99 (6.3×) | 1.39 (4.2×) | 1.52 (3.9×) |

in the last layer, making Primal. a more efficient model. We note that our Primal. further surpasses Flowformer in most cases regarding both running time and memory.

**Long-Range Arena Benchmark**   Long-Range Arena (LRA) [39] is a benchmark for the long-sequence scenarios, including equation calculation (ListOps) [40], review classification (Text) [41], document retrieval (Retrieval) [42], image classification (Image) [43] and image spatial dependencies (Pathfinder) [44]. We follow the settings in [45] with PyTorch. The Transformer backbone is set with 2 layers, hidden dimension 128 with 2 heads and embedding dimension 64 with mean pooling, where Reformer (Re.) uses 2 hashes, Performer (Per.) has 256 random feature dimensions and Linformer (Lin.) uses a projection dimension of 256. Our hyper-parameter is set from $\eta \in \{0.05, 0.1\}$, $s \in \{20, 30\}$. Experiments are conducted on a single NVIDIA Tesla V100 SXM2 32GB GPU.

From the reported top-1 test accuracy in Table 3, our PrimalFormer shows better accuracy while achieves top efficiency (see Table 4) than several efficient self-attention counterparts including Reformer, Performer and Linformer. Notably, our model Primal.+Trans. achieves the state-of-the-art accuracy of 60.4%, which is 1.6% higher than Transformer, and 1.2% higher than the currently best YOSO-E, showing that the deployed Primal-Attention is able to boost the performance upon canonical Transformer distinctively. On top of it, Primal.+Trans. has distinctively higher efficiency and require much less memory than the canonical Transformer as in Table 4, and our Primal. outperforms all compared variants of efficient Transformer. Compared to Table 1 with simpler data, Primal.+Trans. further achieves performance gain over PrimalFormer. The reason can be that the shallower (the first) layer needs more model capacity for depicting the data patterns in the learning, while the deep (the second) layer captures less detailed features and is regularized with more informative feature learning through our Primal-Attention. The efficiency of our Primal-Attention are further pronounced with longer sequences than that of the UEA benchmark in Table 2.

**Reinforcement Learning**   We consider the offline reinforcement learning (RL) performance of our methods on D4RL benchmark [47] designed for continuous control tasks. We choose three different environments controlling the robot movement: HalfCheetah, Hopper and Walker. Our experiments are conducted on three datasets pre-collected under different policies: Medium-Expert, Medium and Medium-Replay. We follow the experimental settings in Flowformer [11], and also compare to Decision Transformer (DT) [10] which is commonly considered as the baseline with state-of-the-art performances based on the canonical attention. Note that offline RL is an auto-regressive task and our Primal-Attention is adapted to its causal version, as described in the Supplementary Material. We adopt the architecture of 3 layers, hidden dimension 256 with 4 heads, and the embedding dimension 64. Our hyper-parameters are set as $\eta = 0.05$, $s \in \{32, 64, 96\}$. Each experiment is run with 3 different seeds on one NVIDIA Tesla P100 SXM2 16GB GPU.

Table 5: Rewards on D4RL [47] datasets. We report mean and variance for 3 seeds. A higher reward and a lower deviation indicates better performance. We consider Decision Transformer (DT), Linear Transformer (Linear.), Reformer (Re.), Performer (Per.), cosFormer (Cos.) and Flowformer (Flow.).

| Dataset | Environment | DT [10] | Linear. [33] | Re. [34] | Per. [14] | Cos. [37] | Flow. [11] | **Ours** Primal.+DT |
|---|---|---|---|---|---|---|---|---|
| Medium -Expert | HalfCheetah | 83.8±3.3 | 78.2±3.2 | 81.5±1.6 | 85.1±2.1 | 85.5±2.9 | 90.8±0.4 | 77.8±22.1 |
| | Hopper | 104.0±2.5 | 107.2±0.9 | 104.2±9.8 | 93.5±13.9 | 98.1±7.4 | 109.9±1.0 | 111.5±0.2 |
| | Walker | 107.7±0.6 | 67.2±27.3 | 71.4±1.8 | 72.6±2.4 | 100.5±14.5 | 108.0±0.4 | 108.9±0.1 |
| Medium | HalfCheetah | 42.4±0.1 | 42.3±0.2 | 42.2±0.1 | 42.1±0.2 | 42.1±0.3 | 42.2±0.2 | 43.0±0.0 |
| | Hopper | 64.2±1.1 | 58.7±0.4 | 59.9±0.7 | 59.7±7.5 | 59.8±3.8 | 66.9±2.5 | 74.5±0.6 |
| | Walker | 70.6±3.2 | 57.9±10.6 | 65.8±4.9 | 63.3±10.7 | 71.4±1.2 | 71.7±2.5 | 77.9±7.8 |
| Medium -Replay | HalfCheetah | 34.6±0.6 | 32.1±1.5 | 33.6±0.7 | 31.7±0.9 | 32.8±3.6 | 34.7±1.5 | 38.9±0.4 |
| | Hopper | 79.7±7.4 | 74.3±7.0 | 66.1±2.6 | 64.6±24.2 | 59.3±16.5 | 75.5±14.5 | 88.5±12.5 |
| | Walker | 62.9±5.0 | 62.1±7.4 | 50.1±3.5 | 61.3±6.7 | 60.5±9.9 | 62.0±3.1 | 76.8±10.3 |
| Average Reward | | 72.2±**2.6** | 64.4±6.5 | 63.9±2.9 | 63.8±7.6 | 67.8±7.6 | 73.5±2.9 | **77.5**±6.0 |

Table 6: Efficiency comparisons on running time (s/1K-steps) and memory (GB) on D4RL [47].

| Model | Medium-Expert | | Medium | | Medium-Replay | |
|---|---|---|---|---|---|---|
| | Time | Memory | Time | Memory | Time | Memory |
| DT (average reward: 72.2) | 20.8 | 0.3 | 20.8 | 0.3 | 20.8 | 0.3 |
| Flow. (average reward: 73.5) | 54.4 | 1.5 | 54.4 | 1.5 | 54.3 | 1.5 |
| Primal.+DT (average reward: 77.5) | 23.5 | 0.3 | 23.4 | 0.3 | 23.3 | 0.3 |

Results in Table 5 demonstrate that our Primal.+ outperforms all compared methods with a distinctly higher average reward. Specifically, our Primal.+DT reaches $4.0$ points higher than the state-of-the-art Flowformer [11]. Compared to the baseline Decision Transformer (DT), our Primal.+ only replaces the self-attention in the top layer and keeps other structures the same and manages to improve the average reward by $5.3$ points. This verifies the effectiveness of our Primal-Attention and the benefits of our low-rank regularization of KSVD for the generality of DT in offline reinforcement learning. We evaluate the efficiency with comparisons to DT and Flowformer in Table 6, which give distinctively better results than other baselines in Table 5. Our Primal.+DT achieves comparable time and memory efficiency as the most efficient baseline DT, while Flowformer shows significantly lower efficiency in both aspects. Recall that our Primal.+DT achieves an average reward of 77.5, which is 5.3 points higher than DT and 4.0 points higher than Flowformer.

**Large-scale Experiments** We evaluate the capability of our Primal.+ model with DeiT-Small/16 [7] as backbone on ImageNet-100 [48] and ImageNet-1K [23] for image classification task. We also experiment with the language modelling task on WikiText-103 [49]. On ImageNet, we train DeiT-Small/16 and our Primal.+DeiT-Small/16 from scratch following the same training protocols in [7] with 4 NVIDIA Tesla V100 SXM2 32GB GPUs. Our hyper-parameters are chosen as $\eta = 0.05$, $s \in \{32, 64, 96\}$. On WikiText-103, we follow the setting in [50] where the sequence length is set as 512, the model consists of 6 decoder layers with 8 heads. We implement the causal version of Primal-Attention in the last layer, i.e., Primal.+Trans with $\eta = 0.1$, $s = 30$. Models are trained from scratch on 4 NVIDIA Tesla V100 SXM2 32GB GPUs for 150K updates after 6K-steps warm-up.

Table 7(a) provides the test accuracy, training time and memory on a single V100 GPU with batch size 256 on both ImageNet-100 and ImageNet-1K datasets. There is only one set of time and memory since models follow the same training protocols on both datasets. Our Primal.+ achieves better accuracy than DeiT-Small/16 on ImageNet-100. It also achieves the same accuracy as baseline with less training time and memory on ImageNet-1K. On WikiText-103 in Table 7(b), our method with default setups significantly reduces the perplexity by 2.0 points than Transformer, and achieves comparable performances with the well-tuned Flowformer, a latest SoTA model, with enhanced efficiency. Results show that using Primal-Attention in the last layer helps the overall performance.

**Ablation Study on Using Two Projections** $e(\boldsymbol{x})$, $r(\boldsymbol{x})$ We conduct an ablation study with (w/) and without (w/o) projection scores, i.e., $r$-scores, involving the left singular vectors, on LRA [39]. Table 8 shows that using both projections (w/ $r$-scores) helps boost performances, verifying the effectiveness of learning with the two sides of singular vectors on an asymmetric attention kernel.

Table 7: Results on large-scale experiments including image classification and language modelling.

(a) Test accuracy (%) and efficiency on ImageNet-100 [48] and ImageNet-1K [23].

| Model | ImageNet-100 (Top-1 Acc.) | ImageNet-1K (Top-1 Acc.) | Time (s/1K-steps) | Memory (GB) |
|---|---|---|---|---|
| DeiT-Small/16 | 74.2 | 79.8 | 2425.5 | 14.2 |
| Primal.+DeiT-Small/16 | **75.7** | 79.8 | **2330.2** | **14.0** |

(b) Results on WikiText-103 [49].

| Model | Perplexity ($\downarrow$) | Time (s/1K-steps) | Memory (GB) |
|---|---|---|---|
| Trans. [1] | 33.0 | 3108.4 | 9.0 |
| Flow. [11] | **30.8** | 3998.4 | 10.5 |
| Primal.+Trans. | 31.0 | **3104.0** | **8.9** |

Table 8: Ablation on $r$-scores involving left singular vectors on LRA [39] with test accuracy (%).

| Model | $r$-scores | ListOps | Text | Retrieval | Image | Pathfinder | Avg. Acc |
|---|---|---|---|---|---|---|---|
| Primal. | w/o | 36.8 | 52.4 | 58.2 | 30.5 | 50.2 | 45.6 |
| | w/ | **37.3** | **61.2** | **77.8** | **43.0** | **68.3** | **57.5** |
| Primal.+Trans. | w/o | 37.1 | 65.1 | 79.2 | 42.8 | 72.8 | 59.4 |
| | w/ | **37.3** | **65.4** | **81.0** | **43.9** | **74.3** | **60.4** |

# 6 Related work

Since the pioneering work [12], the kernel-based approaches have become popular in Transformers, in which the kernel interpretation on the attention matrix has been shed light on. FourierFormer [17] treats the canonical self-attention as non-parametric regression with methodologies for symmetric kernels. [18] considers relative positional embedding with conditional positive definite kernel. [19] treats self-attention operation as support vector regression without considering the asymmetry in the deployed kernel methods, and the supervised regression is not applied in optimzing the attention either. [51] addresses the issue of asymmetry, however, it resorts to symmetrization by replacing the softmax attention with an approximated symmetric one, thereby still dismissing the asymmetry. These prior works deploy kernel-based techniques that are originally designed for symmetric kernels and request to suffice Mercer conditions, which is inconsistent with the asymmetric nature in self-attention, resulting in a nontrivial gap between the analytical understanding and numerical implementation towards revealing the rationale in Transformers. In [21], it leverages the kernel tricks from RKBS [22] that allows asymmetry and formulates attention as a binary kernel learning problem via empirical risk minimization. However, it is hard to find an explicit optimization accordingly in implementing Transformers. Nevertheless, [21] provides an analytical tool from the aspect of kernel represener theorem upon RKBS that allows asymmetry.

Much literature has also devoted to improving the efficiency of the attention computation through different approximation techniques. In the related works addressing the attention mechanism, Reformer [34] uses locally-sensitive hashing for sparse approximation. Performer [14] approximates self-attention matrix with random features. Linformer [46] considers low-rank approximation with the help of random projections. Nyströmformer [45] utilizes the Nyström method by down sampling the queries and keys in the attention matrix. [52] incorporates sparsity prior on attention. These works pose the focus on reducing the computation of the attention kernel matrix from the canonical self-attention. Hence, these works all address how to solve the problem in the dual form involving the kernel matrix, while we work in a significantly different way, that is, in the primal form.

# 7 Conclusion

In this paper, we interpret the self-attention in Transformers with asymmetric kernels and construct a learning framework with SVD on asymmetric kernels (KSVD) under the setups of LSSVM. Within the context of KSVD, a primal-dual model representation is formulated for self-attention and a novel attention mechanism (Primal-Attention) is proposed accordingly by leveraging the primal representation. It is quite significant that with Primal-Attention, not only the computation of the attention kernel matrix in the dual can be avoided, but also the cast unsupervised KSVD optimization can be efficiently incorporated into the training through an additional regularization loss for more informative low-rank property. The analytical derivations and numerical evaluations demonstrate our great potentials in bridging explicit model interpretability and state-of-the-art performances. Future works can include developing different variants with the low-rank property, e.g., robust Transformers, investigating more general applications of Primal-Attention to a wide range of architectures and tasks.

# Acknowledgements

This work is jointly supported by the European Research Council under the European Union's Horizon 2020 research and innovation program/ERC Advanced Grant E-DUALITY (787960), iBOF project Tensor Tools for Taming the Curse (3E221427), Research Council KU Leuven: Optimization framework for deep kernel machines C14/18/068, KU Leuven Grant CoE PFV/10/002, The Research Foundation–Flanders (FWO) projects: GOA4917N (Deep Restricted kernel Machines: Methods and Foundations), Ph.D./Postdoctoral grant, the Flemish Government (AI Research Program), EU H2020 ICT-48 Network TAILOR (Foundations of Trustworthy AI-Integrating Reasoning, Learning and Optimization), Leuven.AI Institute.

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
