# Supplementary Material
# Primal-Attention: Self-attention through Asymmetric Kernel SVD in Primal Representation

**Yingyi Chen** *
ESAT-STADIUS
KU Leuven, Belgium
yingyi.chen@esat.kuleuven.be

**Qinghua Tao** *
ESAT-STADIUS
KU Leuven, Belgium
qinghua.tao@esat.kuleuven.be

**Francesco Tonin**
ESAT-STADIUS
KU Leuven, Belgium
francesco.tonin@esat.kuleuven.be

**Johan A.K. Suykens**
ESAT-STADIUS
KU Leuven, Belgium
johan.suykens@esat.kuleuven.be

In this material, we present the proofs of all analytical results in the paper and additional comments in Section A. More experimental details and results are also provided in Section B. We also provide the broader impacts of our work in Section C.

## A   Theoretical Proofs

In this section, we provide the proofs of all analytical results presented in the paper, covering Theorem 3.2, Remark 3.3, and Lemma 4.2. Additional comments are also provided following each proof in this material.

### A.1   Proof of Theorem 3.2

*Proof of Theorem 3.2.* Given the matrix $X \in \mathbb{R}^{N \times d}$ consists of sequence data $\{\boldsymbol{x}_i \in \mathbb{R}^d\}_{i=1}^N$, the primal optimization problem in self-attention of KSVD with the constructed data-dependent projection weights is formulated as follows, i.e., (6) in the paper:

$$\begin{aligned}
\max_{W_e, W_r, \boldsymbol{e}_i, \boldsymbol{r}_j} J &= \frac{1}{2} \sum_{i=1}^N \boldsymbol{e}_i^\top \Lambda \boldsymbol{e}_i + \frac{1}{2} \sum_{j=1}^N \boldsymbol{r}_j^\top \Lambda \boldsymbol{r}_j - \operatorname{Tr}\left(W_e^\top W_r\right) \\
\text{s.t.} \quad \boldsymbol{e}_i &= (f(X)^\top W_e)^\top \phi_q(\boldsymbol{x}_i), \ i = 1, \dots, N, \\
\boldsymbol{r}_j &= (f(X)^\top W_r)^\top \phi_k(\boldsymbol{x}_j), \ j = 1, \dots, N,
\end{aligned}$$

(1)

where the projection weights of the feature maps $\phi_q(\cdot), \phi_k(\cdot) : \mathbb{R}^d \to \mathbb{R}^p$ can be further denoted as $f(X)^\top W_e =: W_{e|X} \in \mathbb{R}^{p \times s}$, $f(X)^\top W_r =: W_{r|X} \in \mathbb{R}^{p \times s}$ relying on parameters $W_e, W_r \in \mathbb{R}^{N \times s}$ and the constant transformation matrix $f(X) =: F_X \in \mathbb{R}^{N \times p}$, the regularization coefficient denoted by $\Lambda \in \mathbb{R}^{s \times s}$ is a positive diagonal matrix.

The Lagrangian of (1) is

$$\begin{aligned}
\mathcal{L}(W_e, W_r, \boldsymbol{e}_i, \boldsymbol{r}_j, \boldsymbol{h}_{e_i}, \boldsymbol{h}_{r_j}) &= \frac{1}{2} \sum_{i=1}^N \boldsymbol{e}_i^\top \Lambda \boldsymbol{e}_i + \frac{1}{2} \sum_{j=1}^N \boldsymbol{r}_j^\top \Lambda \boldsymbol{r}_j - \operatorname{Tr}\left(W_e^\top W_r\right) \\
&- \sum_{i=1}^N \boldsymbol{h}_{e_i}^\top \left(\boldsymbol{e}_i - W_e^\top f(X) \phi_q(\boldsymbol{x}_i)\right) - \sum_{j=1}^N \boldsymbol{h}_{r_j}^\top \left(\boldsymbol{r}_j - W_r^\top f(X) \phi_k(\boldsymbol{x}_j)\right),
\end{aligned}$$

(2)

---

*Equal contribution.

37th Conference on Neural Information Processing Systems (NeurIPS 2023).

where two sets of dual variable vectors, i.e., $\boldsymbol{h}_{e_i}, \boldsymbol{h}_{r_j} \in \mathbb{R}^s$, are introduced to the equality constraints regarding the projection scores $\boldsymbol{e}_i$ and $\boldsymbol{r}_j$, for $i, j = 1, \ldots, N$, respectively.

By taking the partial derivatives to the Lagrangian (2), the Karush-Kuhn-Tucker (KKT) conditions then lead to:

$$
\begin{cases}
\dfrac{\partial \mathcal{L}}{\partial W_e} = 0 \implies W_r = \sum_{i=1}^{N} f(X)\phi_q(\boldsymbol{x}_i)\boldsymbol{h}_{e_i}^{\top}, \\[2mm]
\dfrac{\partial \mathcal{L}}{\partial W_r} = 0 \implies W_e = \sum_{j=1}^{N} f(X)\phi_k(\boldsymbol{x}_j)\boldsymbol{h}_{r_j}^{\top} \\[2mm]
\dfrac{\partial \mathcal{L}}{\partial \boldsymbol{e}_i} = 0 \implies \Lambda \boldsymbol{e}_i = \boldsymbol{h}_{e_i}, \, i = 1, \ldots, N, \\[2mm]
\dfrac{\partial \mathcal{L}}{\partial \boldsymbol{r}_j} = 0 \implies \Lambda \boldsymbol{r}_j = \boldsymbol{h}_{r_j}, \, j = 1, \ldots, N, \\[2mm]
\dfrac{\partial \mathcal{L}}{\partial \boldsymbol{h}_{e_i}} = 0 \implies W_e^{\top} f(X)\phi_q(\boldsymbol{x}_i) = \boldsymbol{e}_i, \, i = 1, \ldots, N, \\[2mm]
\dfrac{\partial \mathcal{L}}{\partial \boldsymbol{h}_{r_j}} = 0 \implies W_r^{\top} f(X)\phi_k(\boldsymbol{x}_j) = \boldsymbol{r}_j, \, i = 1, \ldots, N.
\end{cases}
\tag{3}
$$

By eliminating the primal variables $W_e$, $W_r$ in KKT conditions (3), we then have

$$
\begin{cases}
\sum_{j=1}^{N} \boldsymbol{h}_{r_j}\phi_k(\boldsymbol{x}_j)^{\top} f(X)^{\top} f(X)\phi_q(\boldsymbol{x}_i) = \Lambda^{-1}\boldsymbol{h}_{e_i}, \, i = 1, \ldots, N, \\[2mm]
\sum_{i=1}^{N} \boldsymbol{h}_{e_i}\phi_q(\boldsymbol{x}_i)^{\top} f(X)^{\top} f(X)\phi_k(\boldsymbol{x}_j) = \Lambda^{-1}\boldsymbol{h}_{r_j}, \, j = 1, \ldots, N,
\end{cases}
$$

which can be expressed in the matrix form as

$$
\begin{bmatrix} \mathbf{0}_{N \times N} & \left[\phi_q(\boldsymbol{x}_i)^{\top} f(X)^{\top} f(X)\phi_k(\boldsymbol{x}_j)\right] \\ \left[\phi_k(\boldsymbol{x}_j)^{\top} f(X)^{\top} f(X)\phi_q(\boldsymbol{x}_i)\right] & \mathbf{0}_{N \times N} \end{bmatrix} \begin{bmatrix} H_e \\ H_r \end{bmatrix} = \begin{bmatrix} H_e \\ H_r \end{bmatrix} \Lambda^{-1},
$$

with $H_e = [\boldsymbol{h}_{e_1}, \ldots, \boldsymbol{h}_{e_N}]^{\top} \in \mathbb{R}^{N \times s}$ and $H_r = [\boldsymbol{h}_{r_1}, \ldots, \boldsymbol{h}_{r_N}]^{\top} \in \mathbb{R}^{N \times s}$.

Therefore, the optimization problem of KSVD in the dual yields the following shifted eigenvalue problem with an asymmetric kernel matrix $K$, such that:

$$
\begin{aligned}
KH_r &= H_e\Sigma, \\
K^{\top}H_e &= H_r\Sigma,
\end{aligned}
\tag{4}
$$

which collects the solutions corresponding to the non-zero entries in $\Lambda$ such that $\Sigma \triangleq \Lambda^{-1}$. The asymmetric kernel $K$ contains the entries induced as $K_{ij} := \langle f(X)\phi_q(\boldsymbol{x}_i), f(X)\phi_k(\boldsymbol{x}_j) \rangle =: \langle \phi_q'(\boldsymbol{x}_i), \phi_k'(\boldsymbol{x}_j) \rangle$, $i, j = 1, \ldots, N$. From the Lanczos Decomposition Theorem [29], i.e., Theorem 2.2 in the paper, we can see that the solutions to the dual problem of KSVD in self-attention, i.e., $H_e$ and $H_r$, correspond to the left and right singular vectors of the asymmetric kernel matrix $K$, where $\Sigma$ serves as the corresponding singular values. This completes the proof.

$\square$

**Comments on Theorem 3.2**  With the primal problem in (6) in the paper, Theorem 3.2 provides the corresponding dual problem of KSVD formulated for self-attention. In [27], a novel variational principle is proposed for SVD with LSSVMs, where a primal-dual formulation for the matrix (linear) SVD is derived. Our KSVD leverages the kernel-based learning framework from [27], however, in addition to our specific application of interpreting self-attention, there are other significant differences and non-trivial novelties in our work:

*i)* [27] mainly addresses the original SVD given any data matrix, while we formulate the non-linear extension leading to the asymmetric attention matrix in relation to the queries and keys. Additionally, [27] presents the optimization w.r.t. a single projection direction in the linear SVD, while we generalize the formulation to multiple projection directions in the matrix form.

*ii)* The data sources for the two non-linear feature maps are related to the queries and keys, as opposed to [27] that specifies the two data sources as the rows and columns of the given data matrix. Therefore, our KSVD is more general in the data setups.

*iii)* Rather than using only the data-independent projection weights $W_e, W_r$ as linear mappings in [27], we propose the generalized form that allows extra transformation matrix dependent non-linearly on the sequence data for self-attention.

In particular, the benefits and motivations of our data-dependent projection weights are as follows:

*i)* In the canonical self-attention, the values vary for different input sequence data, and later in Remark 3.3, we show that the values can be regarded as playing the role of the dual variables in KSVD. Inspired by this property, we introduce input sequence data information to the corresponding primal variables.

*ii)* In the proposed Primal-Attention, the data-dependent projection weights provide more degrees of freedom to improve model's representation ability.

*iii)* Using data-dependent projection weights does not affect the derivation of the shifted eigenvalue problem in the dual. Specifically, when the transformation matrix $F_X$ is chosen as an identity matrix for a simpler structure, it boils downs to the data-independent case, where the kernel $K$ in self-attention is obtained with entries $K_{ij} = \langle \phi_q(\boldsymbol{x}_i), \phi_k(\boldsymbol{x}_j) \rangle$, $i, j = 1, \ldots, N$.

Provided with the general form of the projections weights in (1), practitioners can flexibly adjust the KSVD setups for the self-attention implementation. Related empirical studies can be referred to Section B.2 in this material.

## A.2    Proof of Remark 3.3

With the derivations of the primal-dual optimization problems above, the primal-dual model representation of our KSVD problem can be provided correspondingly. The proof of Remark 3.3 in the paper closely follows the proof of Theorem 3.2, and we show it as follows.

*Proof of Remark 3.3.* The primal model representations for the self-attention outputs in (1) are

$$\text{Primal:} \quad \begin{cases} e(\boldsymbol{x}) = (f(X)^\top W_e)^\top \phi_q(\boldsymbol{x}), \\ r(\boldsymbol{x}) = (f(X)^\top W_r)^\top \phi_k(\boldsymbol{x}). \end{cases} \tag{5}$$

The dual model representations for the self-attention outputs can be derived by eliminating the primal variables with (3):

$$\text{Dual:} \quad \begin{cases} e(\boldsymbol{x}) = W_e^\top f(X) \phi_q(\boldsymbol{x}) = \big( \sum_{j=1}^N f(X) \phi_k(\boldsymbol{x}_j) \boldsymbol{h}_{r_j}^\top \big)^\top f(X) \phi_q(\boldsymbol{x}) \\ \qquad\quad = \sum_{j=1}^N \boldsymbol{h}_{r_j} \phi_q(\boldsymbol{x})^\top f(X)^\top f(X) \phi_k(\boldsymbol{x_j}), \\ r(\boldsymbol{x}) = W_r^\top f(X) \phi_k(\boldsymbol{x}) = \big( \sum_{i=1}^N f(X) \phi_q(\boldsymbol{x}_i) \boldsymbol{h}_{e_i}^\top \big)^\top f(X) \phi_k(\boldsymbol{x}) \\ \qquad\quad = \sum_{i=1}^N \boldsymbol{h}_{e_i} \phi_q(\boldsymbol{x_i})^\top f(X)^\top f(X) \phi_k(\boldsymbol{x}). \end{cases} \tag{6}$$

Further, with the kernel trick in the dual optimization problem (4), i.e.,

$$\kappa(\boldsymbol{x}_i, \boldsymbol{x}_j) := \langle f(X) \phi_q(\boldsymbol{x}_i), f(X) \phi_k(\boldsymbol{x}_j) \rangle, \ i, j = 1, \ldots, N,$$

we then attain the primal-dual representations of KSVD that allows data-dependent projection weights for self-attention as follows:

$$\text{Primal:} \quad \begin{cases} e(\boldsymbol{x}) = W_{e|X}^\top \phi_q(\boldsymbol{x}) \\ r(\boldsymbol{x}) = W_{r|X}^\top \phi_k(\boldsymbol{x}) \end{cases}, \quad \text{Dual:} \quad \begin{cases} e(\boldsymbol{x}) = \sum_{j=1}^N \boldsymbol{h}_{r_j} \kappa(\boldsymbol{x}, \boldsymbol{x}_j) \\ r(\boldsymbol{x}) = \sum_{i=1}^N \boldsymbol{h}_{e_i} \kappa(\boldsymbol{x}_i, \boldsymbol{x}) \end{cases},$$

where $W_{e|X} := f(X)^\top W_e \in \mathbb{R}^{p \times s}$, $W_{r|X} := f(X)^\top W_r \in \mathbb{R}^{p \times s}$. $\qquad\square$

**Comments on Remark 3.3** With Remark 3.3, we can equivalently represent the projection scores in different ways, i.e., either through the feature maps in the primal or the kernel matrix in the dual. Under the framework of KSVD, the existing attention outputs can be interpreted as the projection scores $e(\boldsymbol{x})$ in the dual representation, where the values correspond to the dual variables $\boldsymbol{h}_{r_j}$. The primal-dual models provide versatile alternatives to represent and understand the attention outputs. Notably, the primal representation can avoid the computation of the kernel matrix which is widely considered as an obstacle to the computational efficiency of attention. In addition, we find that there exists another set of projections reflecting the asymmetry information, i.e., $r(\boldsymbol{x})$. Motivated by the above, we propose our new self-attention mechanism, i.e., Primal-Attention in Section 4 in the paper.

### A.3 Proof of Lemma 4.2

Lemma 4.2 evaluates the objective value $J$ in the primal optimization problem (1) when the solutions satisfy the stationarity conditions in (4).

*Proof of Lemma 4.2.* Based on the KKT conditions in (3), by eliminating the primal variables $W_e, W_r, \boldsymbol{e}_i, \boldsymbol{r}_j$, the optimization objective $J$ is given by

$$
\begin{aligned}
J &= \frac{1}{2}\sum_{i=1}^{N} \boldsymbol{e}_i^\top \Lambda \boldsymbol{e}_i + \frac{1}{2}\sum_{j=1}^{N} \boldsymbol{r}_j^\top \Lambda \boldsymbol{r}_j - \mathrm{Tr}\left(W_e^\top W_r\right) \\
&= \frac{1}{2}\sum_{i=1}^{N} \boldsymbol{e}_i^\top \Lambda \boldsymbol{e}_i + \frac{1}{2}\sum_{j=1}^{N} \boldsymbol{r}_j^\top \Lambda \boldsymbol{r}_j - \mathrm{Tr}\left(W_r^\top W_e\right) \\
&= \frac{1}{2}\sum_{i=1}^{N} (\Lambda^{-1}\boldsymbol{h}_{e_i})^\top \Lambda \Lambda^{-1}\boldsymbol{h}_{e_i} + \frac{1}{2}\sum_{j=1}^{N} (\Lambda^{-1}\boldsymbol{h}_{r_j})^\top \Lambda \Lambda^{-1}\boldsymbol{h}_{r_j} \\
&\quad - \mathrm{Tr}\left(\left(\sum_{i=1}^{N} f(X)\phi_q(\boldsymbol{x}_i)\boldsymbol{h}_{e_i}^\top\right)^\top \sum_{j=1}^{N} f(X)\phi_k(\boldsymbol{x}_j)\boldsymbol{h}_{r_j}^\top\right) \\
&= \frac{1}{2}\sum_{i=1}^{N} \boldsymbol{h}_{e_i}^\top \Lambda^{-1}\boldsymbol{h}_{e_i} + \frac{1}{2}\sum_{j=1}^{N} \boldsymbol{h}_{r_j}^\top \Lambda^{-1}\boldsymbol{h}_{r_j} \qquad (7) \\
&\quad - \mathrm{Tr}\left(\left(\sum_{i=1}^{N} f(X)\phi_q(\boldsymbol{x}_i)\boldsymbol{h}_{e_i}^\top\right)^\top \sum_{j=1}^{N} f(X)\phi_k(\boldsymbol{x}_j)\boldsymbol{h}_{r_j}^\top\right) \\
&= \frac{1}{2}\sum_{i=1}^{N} \boldsymbol{h}_{e_i}^\top \Lambda^{-1}\boldsymbol{h}_{e_i} + \frac{1}{2}\sum_{j=1}^{N} \boldsymbol{h}_{r_j}^\top \Lambda^{-1}\boldsymbol{h}_{r_j} - \mathrm{Tr}\left(H_e^\top K H_r\right) \\
&\overset{(4)}{=} \frac{1}{2}\sum_{i=1}^{N} \boldsymbol{h}_{e_i}^\top \Sigma \boldsymbol{h}_{e_i} + \frac{1}{2}\sum_{j=1}^{N} \boldsymbol{h}_{r_j}^\top \Sigma \boldsymbol{h}_{r_j} - \mathrm{Tr}\left(H_e^\top H_e \Sigma\right) \\
&= \frac{1}{2}\sum_{l=1}^{s} \sigma_l \boldsymbol{h}_{e,l}^\top \boldsymbol{h}_{e,l} + \frac{1}{2}\sum_{l=1}^{s} \sigma_l \boldsymbol{h}_{r,l}^\top \boldsymbol{h}_{r,l} - \sum_{l=1}^{s} \sigma_l \boldsymbol{h}_{e,l}^\top \boldsymbol{h}_{e,l},
\end{aligned}
$$

where in the last equation, we denote the dual variables corresponding to the $l$-th projection direction, i.e., singular vectors in relation to the singular value $\sigma_l$, as $\boldsymbol{h}_{e,l} := H_e[:,l] = [\boldsymbol{h}_{e_1}[l], \ldots, \boldsymbol{h}_{e_N}[l]]^\top \in \mathbb{R}^N$, $\boldsymbol{h}_{r,l} := H_r[:,l] = [\boldsymbol{h}_{r_1}[l], \ldots, \boldsymbol{h}_{r_N}[l]]^\top \in \mathbb{R}^N$, and $\Sigma = \mathrm{diag}\{\sigma_1, \ldots, \sigma_s\} \triangleq \Lambda^{-1}$.

Based on both (7) and Theorem 3.2 in the paper, we have the following equations:

$$
\begin{aligned}
K^\top K \boldsymbol{h}_{r,l} &= \sigma_l K^\top \boldsymbol{h}_{e,l} = \sigma_l^2 \boldsymbol{h}_{r,l}, \\
K K^\top \boldsymbol{h}_{e,l} &= \sigma_l K \boldsymbol{h}_{r,l} = \sigma_l^2 \boldsymbol{h}_{e,l}.
\end{aligned}
$$

Hence, we can rewrite

$$
\begin{aligned}
\boldsymbol{h}_{e,l}^\top \boldsymbol{h}_{e,l} &= \frac{1}{\sigma_l^2}(K K^\top \boldsymbol{h}_{e,l})^\top \boldsymbol{h}_{e,l} = \frac{1}{\sigma_l^2}\boldsymbol{h}_{e,l}^\top K K^\top \boldsymbol{h}_{e,l} = \frac{1}{\sigma_l^2}\boldsymbol{h}_{e,l}^\top (K K^\top \boldsymbol{h}_{e,l}) \\
&= \frac{1}{\sigma_l^2}\boldsymbol{h}_{e,l}^\top (\sigma_l K \boldsymbol{h}_{r,l}) = \frac{1}{\sigma_l}\boldsymbol{h}_{e,l}^\top K \boldsymbol{h}_{r,l} = \frac{1}{\sigma_l}(K^\top \boldsymbol{h}_{e,l})^\top \boldsymbol{h}_{r,l} \qquad (8) \\
&= \boldsymbol{h}_{r,l}^\top \boldsymbol{h}_{r,l},
\end{aligned}
$$

which leads to

$$J = \frac{1}{2}\sum_{l=1}^{s}\sigma_l \boldsymbol{h}_{e,l}^\top \boldsymbol{h}_{e,l} + \frac{1}{2}\sum_{l=1}^{s}\sigma_l \boldsymbol{h}_{r,l}^\top \boldsymbol{h}_{r,l} - \sum_{l=1}^{s}\sigma_l \boldsymbol{h}_{e,l}^\top \boldsymbol{h}_{e,l}$$
$$\stackrel{(8)}{=} \frac{1}{2}\sum_{l=1}^{s}\sigma_l \boldsymbol{h}_{e,l}^\top \boldsymbol{h}_{e,l} + \frac{1}{2}\sum_{l=1}^{s}\sigma_l \boldsymbol{h}_{e,l}^\top \boldsymbol{h}_{e,l} - \sum_{l=1}^{s}\sigma_l \boldsymbol{h}_{e,l}^\top \boldsymbol{h}_{e,l} \qquad (9)$$
$$= 0.$$

Note that (7), (8) and (9) also hold for the data-independent projection weights case where $f(X)$ is an identity matrix. In this case, the entries in the induced asymmetric kernel $K$ become $K_{ij} = \langle f(X)\phi_q(\boldsymbol{x}_i), f(X)\phi_k(\boldsymbol{x}_j)\rangle = \langle \phi_q(\boldsymbol{x}_i), \phi_k(\boldsymbol{x}_j)\rangle$, $i, j = 1, \ldots, N$. This completes the proof. $\square$

**Comments on Lemma 4.2**   With Lemma 4.2, we validate that the objective $J$ (1) in the primal optimization problem reaches zero when the stationarity conditions are satisfied, i.e., the singular vectors and their corresponding singular values in (4) are obtained. In the paper, the KSVD optimization for self-attention is realized by incorporating the objective $J$ as an additional regularization loss to the original task-oriented loss, and then minimizing the total loss to zero as shown in (10) and (11) in the paper. In this manner, we avoid solving the dual optimization that involves a SVD problem on a kernel matrix. Moreover, as in the proof of Theorem 3.2, we note that the regularization coefficient $\Lambda$ in the primal optimization (1) corresponds to the singular values in the dual optimization (4). With the SGD-based or AdamW-based algorithm, we flexibly integrate the hyper-parameter selection of $\Lambda$ into the optimization by setting $\Lambda$ as a learnable parameter. In this case, $\Lambda$ can be optimized together with other model parameters by simply minimizing the total loss in (10) in the paper.

# B   More Experimental Results

## B.1   Setup Details

This section provides the implementation details of all experiments included in the paper. Firstly, we outline the main algorithm of our Primal-Attention mechanism in Algorithm 1 for clarity. Note that in data-dependent cases with $f(X)$, we take a subset of $X$ by uniformly sampling $n$ points from $X$ for efficiency aspects, as the main patterns of a matrix can be retained with random linear projections shown by the Johnson–Lindenstrauss lemma [30]. This will be illustrated in details in the following.

---

**Algorithm 1** Learning with Primal-Attention

---

**Require:** $X := [\boldsymbol{x}_1, \ldots, \boldsymbol{x}_N]^\top \in \mathbb{R}^{N \times d}$ is the input sequence to the attention block in Transformer, mappings $g_q(\cdot): \mathbb{R}^{d_q} \to \mathbb{R}^p$, $g_k(\cdot): \mathbb{R}^{d_k} \to \mathbb{R}^p$ defined in (6) in the paper, number of projection directions $s$ defined in (6) in the paper, regularization coefficient $\eta$ defined in (10) in the paper,

**Ensure:** Transformation matrix $f(X) =: F_X \in \mathbb{R}^{N \times p}$ defined in (6) in the paper is required if data-dependent projection weights are used.

    **if** Data-dependent projection weights **then**

        $q(\boldsymbol{x}_i) = W_q \boldsymbol{x}_i, k(\boldsymbol{x}_i) = W_k \boldsymbol{x}_i;$                           $\triangleright W_q \in \mathbb{R}^{d_q \times d}, W_k \in \mathbb{R}^{d_k \times d}$

        $e(\boldsymbol{x}_i) = (f(X)^\top W_e)^\top g_q(q(\boldsymbol{x}_i));$             $\triangleright$ compute $e$-score for $i = 1, \ldots, N$

        $r(\boldsymbol{x}_i) = (f(X)^\top W_r)^\top g_k(k(\boldsymbol{x}_i));$             $\triangleright$ compute $r$-score for $i = 1, \ldots, N$

        $\boldsymbol{o}_i = W_o[e(\boldsymbol{x}_i); r(\boldsymbol{x}_i)];$        $\triangleright$ compute concatenated output with $W_o \in \mathbb{R}^{d_v \times (2s)}$

    **else if** Data-independent projection weights **then**

        $q(\boldsymbol{x}_i) = W_q \boldsymbol{x}_i, k(\boldsymbol{x}_i) = W_k \boldsymbol{x}_i;$                           $\triangleright W_q \in \mathbb{R}^{d_q \times d}, W_k \in \mathbb{R}^{d_k \times d}$

        $e(\boldsymbol{x}_i) = W_e^\top g_q(q(\boldsymbol{x}_i));$          $\triangleright W_e \in \mathbb{R}^{p \times s}$, compute $e$-score for $i = 1, \ldots, N$

        $r(\boldsymbol{x}_i) = W_r^\top g_k(k(\boldsymbol{x}_i));$          $\triangleright W_r \in \mathbb{R}^{p \times s}$, compute $r$-score for $i = 1, \ldots, N$

        $\boldsymbol{o}_i = W_o[e(\boldsymbol{x}_i); r(\boldsymbol{x}_i)];$        $\triangleright$ compute concatenated output with $W_o \in \mathbb{R}^{d_v \times (2s)}$

    **end if**

---

**UEA Time Series**   The UEA time series benchmark [31] consists of 30 datasets. Following the setup in [11], we select 10 datasets for evaluation. For all experiments of our PrimalFormer and Primal.+Trans., we adopt the data-dependent projection weights for Primal-Attention, i.e., we have $W_{e|X} := f(X)^\top W_e \in \mathbb{R}^{p \times s}$ and $W_{r|X} := f(X)^\top W_r \in \mathbb{R}^{p \times s}$. On account that some datasets consist of long sequence samples, e.g., EthanolConcentration of length 1751, SelfRegulationSCP1 of

Table 1: Ablation study on the two main hyper-parameters $\eta$ and $s$. We report test accuracy (%) of PrimalFormer on the UEA time series classification archive benchmark [31].

| Dataset | $s$ | $\eta$ 0 | 0.1 | 0.2 | 0.5 | Dataset | $s$ | $\eta$ 0 | 0.1 | 0.2 | 0.5 |
|---|---|---|---|---|---|---|---|---|---|---|---|
| EthanolConcentration | 20 | 32.3 | 31.6 | 30.8 | 32.7 | FaceDetection | 20 | 64.2 | **67.1** | 66.2 | 66.4 |
| | 30 | 30.0 | **33.1** | 31.9 | 30.4 | | 30 | 65.0 | 65.3 | 65.2 | 65.7 |
| | 40 | 30.8 | 32.3 | **33.1** | 31.9 | | 40 | 64.5 | 65.6 | 66.5 | 66.7 |
| HandWriting | 20 | 26.7 | 28.4 | 27.3 | 26.9 | HeartBeat | 20 | 75.1 | 72.7 | 74.2 | 75.1 |
| | 30 | 28.2 | 26.9 | **29.6** | 25.9 | | 30 | 75.6 | 75.6 | 75.6 | **76.1** |
| | 40 | 26.0 | 26.5 | 27.7 | 27.5 | | 40 | **76.1** | 75.1 | 72.2 | 74.6 |
| JapaneseVowels | 20 | 98.1 | 98.1 | 97.3 | 97.8 | PEMS-SF | 20 | 86.1 | 85.6 | 83.8 | 84.4 |
| | 30 | 98.1 | 97.6 | 97.3 | 97.6 | | 30 | 83.8 | 86.7 | 82.1 | 86.7 |
| | 40 | 98.1 | 98.1 | 97.6 | **98.4** | | 40 | 86.1 | **89.6** | 86.7 | 85.0 |
| SelfRegulationSCP1 | 20 | 91.5 | **92.5** | 90.8 | 91.8 | SelfRegulationSCP2 | 20 | **57.2** | 53.9 | 55.6 | 56.1 |
| | 30 | 92.2 | 91.8 | 92.2 | **92.5** | | 30 | 55.6 | 53.9 | 55.6 | 55.8 |
| | 40 | 91.8 | 91.8 | 91.5 | 91.8 | | 40 | 52.9 | 56.1 | 53.9 | 53.3 |
| SpokenArabicDigits | 20 | 100 | 100 | 100 | 100 | UWaveGestureLibrary | 20 | **86.3** | 83.8 | 83.8 | 84.7 |
| | 30 | 100 | 100 | 100 | 100 | | 30 | 85.3 | 85.0 | 85.0 | 84.1 |
| | 40 | 100 | 100 | 100 | 100 | | 40 | **86.3** | 84.1 | 84.1 | 85.9 |

length 896, SelfRegulationSCP2 of length 1152, our choice of $f(X)$ should include more information about $X$ for greater model flexibility while maintaining computational efficiency. In this regard, we set $f(X) := X'$ where $X' \in \mathbb{R}^{n \times p}$ is a subset of the sequence data $X \in \mathbb{R}^{N \times d}$ by uniformly sampling $n = \min\{s * \texttt{rank\_multi}, N\}$ points (rows) from $X$. We set $\texttt{rank\_multi} = 10$ for most cases, and set $\texttt{rank\_multi} = 5$ for datasets including FaceDetection, HandWriting, JapaneseVowels, PEMS-SF and SpokenArabicDigits, since they have shorter sequence lengths. In this manner, the size of the transformation matrix $f(X)$ is implemented as $\mathbb{R}^{n \times p}$ with $n \ll N$, reducing memory requirements especially for long sequence data with large $N$.

**Long-Range Arena**  Long-Range Arena (LRA) [39] consists of long-sequence scenarios: ListOps of 2K sequence length, Text of 4K length, Retrieval of 4K, Image of 1K and Pathfinder of 1K. With joint consideration of performance and efficiency, for all experiments of our PrimalFormer and Primal.+Trans. in the paper, we adopt Primal-Attention with data-dependent projection weights and set $n = \min\{s * 10, N\}$, i.e., $\texttt{rank\_multi} = 10$, for all cases.

**Reinforcement Learning**  D4RL [47] is a suite of continuous control tasks and datasets for benchmarking progress in offline reinforcement learning. In this experiment, we consider Primal.+DT with Decision Transformer (DT) [10] as the backbone. Specifically, we consider a three-layer DT with its self-attention in third layer replaced by our Primal-Attention. As DT utilizes a causal self-attention mask which predicts actions autoregressively, to align with the causal structure, we propose the causal-version of Primal-Attention, namely, Causal Primal-Attention. For clarity, we attach the corresponding PyTorch-like pseudo code in Listing 1 in this material. Note that for this task, we utilize Causal Primal-Attention with the simpler data-independent projection weights, i.e., $W_e, W_r \in \mathbb{R}^{p \times s}$, which helps to prevent overfitting in learning rewards in the RL training process.

**Image Classification**  ImageNet-100 [48] contains 100 classes of images from ImageNet-1K [23]. On both ImageNet-100 and ImageNet-1K, we use Primal.+DeiT-Small/16 with standard DeiT-Small/16 as the backbone. Specifically, the self-attention of the last layer of DeiT-Small/16 is replaced by our Primal-Attention using data-dependent projection weights with the setup $n = \min\{s * 10, N\}$, i.e., $\texttt{rank\_multi} = 10$.

**Language Modelling**  We conduct the language modelling on the WikiText-103 [49], which aims to estimate the probability distribution of a token given the previous ones. We replace the self-attention in the last layer of the Transformer baseline with our Causal Primal-Attention using data-dependent projection weights with the setup $n = \min\{s * 10, N\}$, i.e., $\texttt{rank\_multi} = 10$.

**Listing 1** PyTorch-like Pseudo Code of Causal Primal-Attention for RL

```python
import torch
import torch.nn as nn
import torch.nn.functional as F

class CausalPrimalAttention(nn.Module):
    def __init__(self, d_model, n_heads, low_rank, drop_out):
        super(CausalPrimalAttention, self).__init__()
        self.d_keys = d_model // n_heads # key dimension

        self.query_projection = nn.Linear(d_model, self.d_keys * n_heads) # q(X)
        self.key_projection = nn.Linear(d_model, self.d_keys * n_heads) # k(X)
        self.out_projection = nn.Linear(self.d_keys * n_heads, d_model)
        self.n_heads = n_heads
        self.dropout = nn.Dropout(drop_out)

        # data-independent projection weights
        # "low_rank" is the number of projection directions "s" of KSVD
        self.We = nn.Parameter(nn.init.orthogonal_(torch.Tensor(self.n_heads, self.d_keys, low_rank)))
        self.Wr = nn.Parameter(nn.init.orthogonal_(torch.Tensor(self.n_heads, self.d_keys, low_rank)))
        # projection after concatenating [e-score, r-score]
        self.concate_weight = nn.Linear(2*low_rank, self.d_keys)

    def feature_map(self, x):
        # our cosine similarity kernel related feature maps
        return F.normalize(x, p=2, dim=-1)

    def forward(self, queries, keys, attention_mask=None):
        B, L, _ = queries.shape # batch, length
        _, S, _ = keys.shape # length

        queries = self.query_projection(queries).view(B, L, self.n_heads, -1)
        keys = self.key_projection(keys).view(B, S, self.n_heads, -1)

        # transpose the queries and keys
        queries = queries.transpose(1, 2)
        keys = keys.transpose(1, 2)

        # causal mechanism
        # generate weights for Primal-Attention
        normal = (((torch.arange(L)).float() + 1.0)).to(queries.device)
        # conduct cumsum before the non-linear map (causal map)
        queries = queries.cumsum(dim=2) / normal[None, None, :, None]
        keys = keys.cumsum(dim=2) / normal[None, None, :, None]
        # feature maps
        queries = self.feature_map(queries)
        keys = self.feature_map(keys)
        # compute e-score and r-score
        escore = torch.einsum('...nd,...de->...ne', queries, self.We.unsqueeze(0))
        rscore = torch.einsum('...nd,...de->...ne', keys, self.Wr.unsqueeze(0))
        score = torch.cat((escore, rscore), dim=-1)
        out = self.concate_weight(score).transpose(1, 2).contiguous()
        # final projection
        out = out.reshape(B, L, -1)
        out = self.out_projection(out)
        out = self.dropout(out)
        return out
```

## B.2 Further Ablation Studies

**Ablation on $\eta$ and $s$** The numerical investigations are conducted on the two main hyper-parameters of our Primal-Attention, i.e., the coefficient $\eta$ of the KSVD regularization loss and the number of projection directions $s$. We consider the UEA time series datasets. The results of PrimalFormer, i.e., two-layer Transformer with Primal-Attentions, are given in Table 1 in this material. *Firstly*, compared to $\eta = 0$, a rough tuning of $\eta > 0$ improves the performance for most of the datasets. For example, $\eta > 0$ on FaceDetection leads to consistent improvement over $\eta = 0$. This indicates that the KSVD optimization through the regularization loss $J$ in (11) in the paper indeed brings performance boost over its non-regularized counterpart. *Secondly*, even without the KSVD optimization, i.e., $\eta = 0$, our Primal-Attention already leads to good performance, such as the results on SpokenArabicDigits, SelfRegulationSCP2 with $s = 20$, and UWaveGestureLibrary with $s = 20, 40$. This verifies that the new representation in Primal-Attention in (8) in the paper can effectively represent the self-attention and conduct effective learning in the attention outputs. *Thirdly*, effective learning features can be captured in less dimensions than the original embedding dimension and a performance boost can be

Table 2: Ablation study on hyper-parameter $s$ with $\eta = 0.05$. We report the rewards of Primal.+DT on the D4RL datasets [47]. A higher reward indicates better performance.

| Dataset / Environment | Medium-Expert | | | Medium | | | Medium-Replay | | |
|---|---|---|---|---|---|---|---|---|---|
| | $s$=32 | $s$=64 | $s$=96 | $s$=32 | $s$=64 | $s$=96 | $s$=32 | $s$=64 | $s$=96 |
| HalfCheetah | 56.9 | 73.1 | **75.6** | 42.9 | **43.1** | 42.8 | **39.5** | 37.9 | 39.3 |
| Hopper | 111.2 | **112.0** | 111.1 | 66.7 | 63.1 | **73.8** | 84.2 | **91.7** | 87.3 |
| Walker | 108.7 | 108.9 | **109.0** | 75.1 | **77.1** | 77.0 | 71.8 | 70.4 | **80.5** |

Table 3: Ablation on data-dependent and data-independent projection weights of our Primal-Attention mechanism. We report the test accuracy (%) of Primal.+Trans. on UEA time series datasets [31].

| Data-dependent | Dataset | | | | | | | | | | Avg. Acc. |
|---|---|---|---|---|---|---|---|---|---|---|---|
| | Ethanol Concen. | Face Detec. | Hand Writ. | Heart Beat | JPN Vowels | PEMS -SF | SelfRegu. SCP1 | SelfRegu. SCP2 | Spoken ArabicDig. | UWave GestureLib. | |
| ✗ | 34.6 | 63.5 | **29.3** | 76.1 | **99.2** | 88.4 | **92.8** | **58.3** | 100 | 87.2 | 72.9 |
| ✓ | **35.4** | **63.8** | 28.7 | **77.1** | 98.9 | **90.2** | 92.5 | 56.1 | 100 | **88.4** | **73.1** |

potentially gained with even fewer dimensions through our formulated KSVD. To be specific, the embedding dimension for each head is 64, and we set $s \in \{20, 30, 40\}$ in the experiments. Recall that the average accuracy of the canonical Transformer is 71.9% in Table 1 in the paper, while our PrimalFormer reaches 73.1%, which is 1.2% higher upon the canonical one. These results hence show that an appropriate compression in the number of projection directions by KSVD could lead to performance improvements when the low-rank property is desired. Note that since the kernel matrix in the dual of our Primal-Attention is of size $K \in \mathbb{R}^{N \times N}$, we limit $s$ up to $N$, i.e., $0 < s \leq N$, as there exists at most $N$ projection directions in the corresponding KSVD. In general, larger $s$ is preferred in more complicated tasks with more sophisticated dependency between samples in the sequence data. For instance, the reward learning in RL is such a case where less information compression is desired. This can be verified by the results given in Table 2 in this material, where the best performance is attained with $s$ as 64 or 96 in almost all cases.

**Projection Weights** We investigate the effects of projection weights in the data-dependent and data-independent cases for Primal-Attention. Tables 3 and 4 in this material present the comparisons between data-dependent and data-independent projection weights used in Primal.+Trans. on UEA time series datasets [31] and also LRA benchmark [39]. On both benchmarks, data-dependent projection weights case surpasses its data-independent counterpart. The reason of these results can be that data-dependent projection weights help increasing the model's representation ability and capturing more informative features from the rather long sequences in these datasets. Furthermore, for the data-dependent case, we set $f(X) := X'$ where $X' \in \mathbb{R}^{n \times p}$ is a subset of sequence data by uniformly sampling $n = \min\{s * \texttt{rank\_multi}, N\}$ points from $X \in \mathbb{R}^{N \times p}$. As shown in Table 1 in this material, for a given $\texttt{rank\_multi}$ in each dataset, the increase of $s$ does not make the results fluctuate much. Similar phenomenon is found on the LRA datasets during our experiments. Therefore, for almost all experiments in the paper, we simply set $n = \min\{s * 10, N\}$ as default. This can also serve as a mild suggestion for practitioners in implementation. We note that data-dependent projection weights are not always in favor. For example, in the RL tasks, model is prone to overfit the learning of rewards during training if we adopt the Primal-Attention with data-dependent projection weights. Hence, we take the data-independent case instead. In the generalized form of the projection weights with Primal-Attention, more possibilities of greater model representation ability are provided to fit various tasks and datasets.

## B.3 Further Remarks on Efficiency

**Efficiency with Primal-Attention in architectures** From our learning scheme in Figure 2 in the paper and the empirical efficiency analysis with Tables 2, 4, 6 and 7 in the paper, we can see that the efficiency gain of the Transformers implemented with Primal-Attention over canonical baselines is influenced by two main factors, *i)* the number of Primal-Attention layers employed in the architecture, i.e., the more the better; *ii)* sequence length of the training data, i.e., the longer the more significant:

Table 4: Ablation on data-dependent and data-independent projection weights of our Primal-Attention mechanism. We report the test accuracy (%) of Primal.+Trans. on the LRA benchmark [39].

| Data-dependent | Dataset | | | | | Average Accuracy |
|---|---|---|---|---|---|---|
| | ListOps | Text | Retrieval | Image | Pathfinder | |
| ✗ | 37.0 | 40.2 | 74.3 | 80.8 | **65.6** | 59.6 |
| ✓ | **37.3** | **43.9** | 74.3 | **81.0** | 65.4 | **60.4** |

Table 5: Architecture of Primal.+ on different datasets.

| Primal.+ | canonical_layer+[primal_layer] | num_head | head_dim |
|---|---|---|---|
| UEA | 1+[1] | 8 | 64 |
| LRA | 1+[1] | 2 | 32 |
| D4RL | 2+[1] | 4 | 64 |
| WikiText-103 | 5+[1] | 8 | 64 |
| ImageNet | 11+[1] | 6 | 64 |

*i)* With deep architectures, the efficiency can be further improved by replacing more layers with our Primal-Attention. Yet, in very deep Transformers, Primal-Attention is not necessarily always superior in performance when being applied to all layers, as the learning in shallow layers may not enjoy the benefits from the low-rank property from KSVD as much as the higher layers do. It would be interesting to explore a more generic implementation setup for Primal-Attention in very deep Transformers, as briefly mentioned in the last paragraph of possible future work in this material.

*ii)* The length of the data sequence, i.e., $N$, is also a key factor influencing the efficiency. By avoiding the computation of the $N \times N$ attention matrix, our Primal-Attention can gain better efficiency on longer-sequence datasets. Although ImageNet-1K is large-scale, currently Transformers treat each image as a sequence of length 197 (with `cls` token), which is actually not too long (even compared to some UEA datasets as shown in Table 2 in the paper). Hence, this is also a reason why our Primal.+DeiT-Small/16 does not improve the efficiency significantly in Table 7(a) in the paper. Similarly in WikiText-103, the data sequence length is 512, which is also not really long, hence the efficiency of our Primal.+Trans. is not always superior under the current setups.

**Efficiency gain of Primal.+ in different Tasks** The efficiency gain of Primal.+ over baseline is more significant on UEA and LRA, as the backbone has only 2 layers, hence replacing one layer makes a difference to the overall architecture. Moreover, UEA and LRA in general have longer training sequence length, which would signalize Primal-Attention's efficiency. In contrast, the backbones on D4RL, WikiText-103 and ImageNet have more layers where canonical self-attention layers are the majority structures in Primal.+ as shown in Table 5 in this material. Besides, the efficiency gain is less significant also due to the shorter training sequence lengths on these datasets.

## C  Broader Impacts

**Societal Impacts** In this work, we provide a new perspective to interpret self-attention through a KSVD problem with asymmetric kernels under the LSSVM framework. Compared to the canonical Transformers, our method is more efficient in tackling long-sequence datasets with our more efficient architectures that avoids the kernel matrix computation and also regularize the model with improved low-rank properties. In this aspect, our method is more energy friendly as it can decrease the power consumption during training.

**Possible Future Works** We introduce a new self-attention mechanism from the primal perspective of the KSVD problem where feature maps are utilized rather than the kernels. We currently work on the feature map corresponding to the Cosine similarity kernel in the paper that achieves state-of-the-art performances on the evaluated benchmarks. For more general setups and applications, different feature maps and backbone architectures can be further investigated. Therefore, it can extend our method to a wider range of tasks and possibly gain better performance under practical scenarios. These can be possible directions for future work.