# OpenReview forum: "Primal-Attention: Self-attention through Asymmetric Kernel SVD in Primal Representation"
_NeurIPS.cc/2023/Conference — NeurIPS 2023 poster_

### Official Review · Reviewer_aphr · 2023-06-25

**Soundness:** 3 good
**Presentation:** 2 fair
**Contribution:** 3 good
**Rating:** 7
**Confidence:** 2

**Summary:**

Self-attention mechanisms serve as pivotal components in the domains of natural language processing and computer vision. Previous kernel function based self-attention variants, which are predicated on Mercer kernels, tend to overlook the inherent asymmetry of the attention matrix in the vanilla self-attention. In response to this oversight, this study introduces a novel approach, termed Primal-Attention. This approach leverages asymmetric Kernel Singular Value Decomposition (KSVD) to facilitate a low-rank approximation of the attention matrix, thereby incorporating an asymmetric kernel attention matrix. The empirical findings substantiate that Primal-Attention functions as a low-rank attention mechanism, exhibiting superior performance while maintaining low time complexity.

**Strengths:**

1. This paper introduces a novel perspective that interprets the self-attention mechanism as a KSVD optimization problem.

**Weaknesses:**

1. The exposition of methodologies in this paper lacks sufficient clarity. For instance, the crucial concept of primal optimization in self-attention with KSVD, as proposed in Equation 6, is introduced abruptly without a detailed discussion of the connection between KSVD and self-attention.
2. Algorithm 1, outlined in Appendix B.1, details the entire process of Primal-Attention and employs random projection. However, the omission of a citation for the Johnson–Lindenstrauss lemma in the paper is inappropriate.

**Questions:**

1. Why should the primal optimization function, as depicted in Equation 6, be considered? On what basis is this function proposed? Furthermore, why should this function be maximized rather than minimized?
2. Why is the formula for Primal-Attention in each head represented by Equation 11? The process of derivation appears to be missing.
3. How do the authors handle the positive diagonal matrix $\mathbf{\Lambda}$? Given that $\mathbf{\Lambda}$ is a key component in Equations 6 and 11, can the authors provide a detailed explanation of the process?

---

> ### Author Rebuttal · Authors · 2023-08-08
>
> We thank the reviewer for the constructive comments and the appreciation on the novelty of our work. We address your concerns point-wisely below.
>
> ***R.1 Concept of primal optimization with KSVD in eq.(6) and connections with self-attention.***
>
> The modeling and optimization of KSVD on self-attention are under the LSSVM setup [1]. We extend its linear version (SVD) to the nonlinear one (KSVD), by starting from the primal optimization eq.(6) till deriving the dual problem  eq.(7) for the primal-dual representation of self-attention in eq.(8). As presented in eq.(3), the attention outputs $o_i=\sum\nolimits_{j=1}^N v(x_j)K_{ij}$ are consistent with the dual representation of KSVD $e_i=\sum\nolimits_{j=1}^N h_{r_j}K_{ij}$, derived in eq.(8). We agree with the reviewer and will add more explanations in an early place before eq.(6). More explanations can also refer to line 115-122 on page 4, and Remarks 3.3-3.4 in paper.
>
> ***R.2 Citation of the Johnson–Lindenstrauss lemma in Algorithm 1.***
>
> In Algorithm 1 with data-dependent projection weights, transformation matrix $f(X)$ is used for projection directions, as explained in line 128-132 on page 4. In experiments, we set $f(X):=X'$ with $X'$ a subset uniformly sampled from the rows of $X$. Indeed, the Johnson–Lindenstrauss Lemma [2] shows the main patterns of a matrix can be retained with the random linear projections. We appreciate the helpful referene and will cite it in paper. Other projection parameters are all optimized by SGD.
>
> ***R.3 Why and on what basis is eq.(6) proposed? Why maximization in eq.(6) but minimization in eq.(10)?***
>
> Eq.(6) is based on the variational principle on SVD under the LSSVM framework [1], but extends the original linear SVD in [1] to the nonlinear KSVD with asymmetric kernels in dual, induced by two feature maps $\phi_q(x),\phi_k(x)$ in primal. The primal objective in eq.(6) maximizes the projection variances of $W_{e|X}^\top\phi_q(x),W_{r|X}^\top\phi_k(x)$ regarding queries and keys. This follows the idea of KSVD, because in dual,the corresponding right and left singular vectors learn the directions with the maximal projection variances, w.r.t. row and column data of the asymmetric kernel matrix $K:=[\left<\phi_q(x_i),\phi_k(x_j)\right>]$. The trace term in eq.(6) regularizes the primal variables $W_e,W_r$ and helps deriving the Lagrange function and the dual optimization eq.(7) via KKT conditions. This is the basis of eq.(6). Details can also refer to Remark 3.1, Theorem 3.2 and its proof in Appendix A.1.
>
> We derived that the primal optimization in eq.(6) can be solved through KKT conditions with stationary solutions in the dual optimization derived in eq.(7). As proved in Lemma 4.2, stationary solutions yield a zero-value objective $J$ in eq.(6), thus we do not need to solve the expensive SVD on the asymmetric kernel matrix $K$ in dual but intead we choose to optimize the objective $J$ in primal, i.e., eq.(6), to a zero value. In implementation of such an optimization, this zero-value objective can be flexibly realized by $\min J^2$ with efficient SGD-based algorithms. Therefore, in training, we adopt $\min L+\eta \sum_{l}J_l^2$, i.e., optimizing the loss $L$ with regularization loss terms $J_l^2$ promoting KSVD optimization. Relevant context can be referred in Theorem 3.2 and Lemma 4.2 in paper and their proofs in Appendix A.
>
> ***R.4 Why is the formula for Primal-Attention in each head represented by eq.(11)?***
>
> In the primal optimization of KSVD for each head in eq.(6), the projection scores $e_i, r_j$ are given in the equality constraints. By replacing $e_i, r_j$ in the objective $J$ of eq.(6), we derive the unconstrained optimization with the objective as in eq.(11), which can be referred to line 124-127 on page 6.
>
> ***R.5 How to handle the positive diagonal matrix $\Lambda$ crucial in eq.(6) and (11)? Detailed explanation of the process?***
>
> + Diagonal matrix $\Lambda$ serves as the inverse of the positive singular values $\Sigma$ of the attention kernel matrix $K$, i.e.,$\Sigma=\Lambda^{-1}$, shown in line 152-156 on page 5. Details can be found in Proof of Theorem 3.2, and Comments on Lemma 4.2 in Appendix.
> + In modeling, as attention outputs are formulated as projection scores along directions corresponding to those singular values (vectors), $\Lambda:= \Sigma^{-1}$ is indeed a key component reflecting the importance of each projection direction, exploring low-rank explanations.
> + In experiments, we assure the positivity of $\Lambda$ by defining a squared value of a diagonal matrix $\Lambda:=C^2$ and set it as learnable parameter in $J$ of eq.(11) for each head:
> $J=\frac{1}{2}\sum\_{i=1}^N \||( W\_{e|X}\Lambda^{\frac{1}{2}})^\top\phi_q(x_i)\||_2^2+\frac{1}{2}\sum\_{j=1}^N \||( W\_{r|X}\Lambda^{\frac{1}{2}})^\top\phi_k(x_j)\||_2^2-\text{Tr}(W_e^\top W_r).$
>
> Note that given $\phi_q, \phi_k$, projection weights $W_{e|X},W_{r|X}\in\mathbb R^{p\times s}$ for the $s$ directions and $\Lambda\in \mathbb R^{s\times s}$ are optimized together by SGD to approach an zero-valued $J$. Here, $\Lambda$ imposes different importance on the $s$ projection directions. Due to the production between $W_{e|X},W_{r|X}$ and $\Lambda$, the optimized results only reflect the final results of $W_{e|X}\Lambda^{\frac{1}{2}},W_{{r|X}}\Lambda^{\frac{1}{2}}$, so the $\Lambda$ optimized by SGD is not necessarily the original inversed singular values. However, in empirical low-rank analysis, we can always compute the exact singular values of the self-attention kernel matrix $K$, as shown in Figure 1 in paper, where our Primal-Attention captures more information with less components than the canonical self-attention.
>
> [1] Suykens, J.A.K. “SVD revisited: A new variational principle, compatible feature maps and nonlinear extensions.” Applied and Computational Harmonic Analysis, 2016.
>
> [2] Lindenstrauss, W., and Johnson J. “Extensions of Lipschitz maps into a Hilbert space.” Contemp. Math, 1984.

---

> > ### Comment · Reviewer_aphr · 2023-08-21
> >
> > Thanks for the authors' response. I have no further questions about this paper.

---

### Official Review · Reviewer_8hph · 2023-07-07

**Soundness:** 4 excellent
**Presentation:** 4 excellent
**Contribution:** 4 excellent
**Rating:** 7
**Confidence:** 5

**Summary:**

This paper proposes a new understanding of self-attention in transformers via asymmetric Kernel Singular Value Decomposition (KSVD). In particular, the authors formulate a primal-dual representation of self-attention for maximizing the projection variances in the attention outputs and then derive a new attention mechanism, namely the Primal-Attention, to avoid direct computation of the kernel matrix. Using KKT conditions, they prove the Primal-Attention can obtain a zero-value objective. Experimental results are provided to justify the advantage of the Primal-Attention.

**Strengths:**

1. The idea of formulating self-attention as solving an KSVD problem is novel and of potentially high-impact.

2. The derivation of self-attention from KSVD given in the paper is elegant and correct.

3. The paper is well-written, easy to follow, and enjoyable to read.

**Weaknesses:**

1.  More large-scale experiments, e.g., on full ImageNet or WikiText103 language modeling, should be conducted.

2. Empirical analysis, such as the efficiency analysis of the Primal-Attention vs. the baselines, should be provided to help provide better understanding of the proposed method.

**Questions:**

1. Can the proposed KSVD framework be used to explain other components in a self-attention unit, such as the residual connection, the layer normalization, and the feedforward network?

**Limitations:**

The authors have not clearly addressed the limitations of their work.

---

> ### Author Rebuttal · Authors · 2023-08-08
>
> We thank the reviewer's high appreciation of our work and the insightful comments, which will be addressed point-wisely below.
>
> **R.1 More large-scale experiments**
>
> As suggested, we test on ImageNet-1K and WikiText-103, both showing our promising potentials. Ours achieves the same accuracy as baseline with less memory; recall that our model also shows enhanced low-rank property as in Fig.1 in paper. On WikiText-103, our method with default setups achieves comparable performances with the well-tuned Flowformer, a latest SoTA model.
>
> **Table 1:** Test Acc. (%) on ImageNet-1K.
> |Model|Top-1 |Memory(GB)|
> |:-:|:-:|:-:|
> |DeiT-Small/16|79.8|14.2|
> |Primal.$+$DeiT-Small/16(ours)|79.8|14.0|
>
> **Table 2:** Perplexity on WikiText-103.
> |Model|Perplexity($\downarrow$)|
> |:-:|:-:|
> |Trans.(2017)|33.0|
> |Re.(2020)|33.6|
> |Per.(2021)|37.5|
> |Cos.(2022)|34.1|
> |Flow. w/o competition(2022)|31.2|
> |Flow. w/o allocation(2022)|32.2|
> |Flow.(2022)|30.8|
> |Primal.$+$Trans.(ours)|31.0|
>
> **R.2 More empirical analyses**
>
> **R.2.1 Efficiency**
>
> LRA is a popular benchmark in evaluating efficiency due to long data sequences. We test on LRA where ours shows better efficiency in running time and memory as in Table 3 in paper.
>
> Below, we provide more efficiency evaluations.
> + *UEA Times Series:* Ours show the best efficiency especially in memory where ours are more efficient with longer sequences, e.g., EthanolConcentration and SelfRegulationSCP2.
> + *D4RL Reinforcement Learning:* Our "Primal.$+$DT" achieves comparable time and memory efficiency as Decision Transformer (DT), while Flowformer is significantly less efficient. Recall that in Table 4 in paper, ours attains a much higher average reward of 77.5 than DT (72.2) and FlowFormer (73.5).
>
> **Table 3:** Running time and memory (GB) on UEA Time Series.
> |UEA benchmark (sequence length)|Trans.|Flow.|Primal.+Trans.|Primal.|
> |:-:|:-:|:-:|:-:|:-:|
> |EthanolConcentration (1751)|4.3|2.4|3.3|2.3|
> |FaceDetection (62)|10.7|9.8|5.7|6.9|
> |HandWriting (152)|0.3|0.3|0.3|0.4|
> |HeartBeat (405)|0.7|0.7|0.7|0.4|
> |JapaneseVowels (26)|0.5|0.6|0.5|0.5|
> |PEMS-SF (144)|0.6|0.7|0.7|0.8|
> |SelfRegulationSCP1 (896)|1.8|1.4|1.6|1.4|
> |SelfRegulationSCP2 (1152)|1.8|1.3|1.6|1.3|
> |SpokenArabicDigits (83)|3.7|4.4|4.0|4.5|
> |UWaveGestureLibrary (315)|0.3|0.3|0.3|0.4|
> |**Avg. Time(s/Epoch)**|2.5|2.2|**1.9**|**1.9**|
>
> ||Trans.|Flow.|Primal.+Trans.|Primal.|
> |:-:|:-:|:-:|:-:|:-:|
> |**Memory**|10.9|2.8|6.5|**2.7**|
>
> **Table 4:** Running time(s/1K-steps) and memory(GB) on D4RL.
> |Time|Medium-Expert|Medium|Medium-Replay|
> |:-:|:-:|:-:|:-:|
> |DT(reward: 72.2)|20.8|20.8|20.8|
> |Flow.(reward: 73.5)|54.4|54.4|54.3|
> |Primal.+DT(**reward:77.5**)|23.5|23.4|23.3|
>
> |Memory|Medium-Expert|Medium|Medium-Replay|
> |:-:|:-:|:-:|:-:|
> |DT|0.3|0.3|0.3|
> |Flow.|1.5|1.5|1.5|
> |Primal.+DT|0.3|0.3|0.3|
>
> **R.2.2 Other empirical analyses**
> + *Spectrum of self-attention kernels:* We plot cumulative explained variances of self-attention kernels on ImageNet-1K in Figure 1 in paper. Our method shows enhanced low-rank property, where more information can be captured within less components than the baseline.
> + *Ablation on $\eta$ and $s$:* Through the KSVD regularization coefficient $\eta$ in eq.(10) and the number of components $s$, we verify the effectiveness of our KSVD optimization in Tables 1, 2 in Sec.B.2 in Supplementary Material.
> + *Ablation on the projection weights:* We also compare the data-dependent and data-independent projections in Tables 3 and 4 in Sec.B.2 in Supplementary Material.
> + *Ablation on projection scores from left singular vectors:* We evaluate the results of using with (w/) and without (w/o) the projection scores ($r$-scores) involving left singular vectors. It shows that using both sets of projections (w/ $r$-scores) helps boost performance in learning asymmetric self-attention kernels.
>
> **Table 5:** Ablation on projection scores, i.e., $r$-scores, involving left singular vectors on LRA.
> |Primal.|ListOps|Text|Retrieval|Image|Pathfinder|Avg. Acc.|
> |:-:|:-:|:-:|:-:|:-:|:-:|:-:|
> |w/o $r$-scores|36.8|52.4|58.2|30.5|50.2|45.6|
> |w/ $r$-scores|37.3|61.2|77.8|43.0|68.3|**57.5**|
>
> |Primal.$+$Trans.|ListOps|Text|Retrieval|Image|Pathfinder|Avg. Acc.|
> |:-:|:-:|:-:|:-:|:-:|:-:|:-:|
> |w/o $r$-scores|37.1|65.1|79.2|42.8|72.8|59.4|
> |w/ $r$-scores|37.3|65.4|81.0|43.9|74.3|**60.4**|
>
> **R.3 Possible work on other components**
>
> Thanks for mentioning these interesting perspectives, which we would like to regard as possible future work. For instance, layer normalization might be in relation to orthonormality of singular vectors; the feedforward network and residual connection could be inspiring to connect with deep kernel machines[1,2]. Nevertheless, the authenticity requires rigorous analyses and experiments before arriving at a conclusion.
>
> **R.4 Limitations**
>
> The limitations can refer to the last paragraph in Supplementary Material.
> + In efficiency evaluations, e.g., on UEA time-series, we can note that our time efficiency gain is less distinctive on shorter-length sequence data than on longer ones. Further promoting the efficiency even on shorter-length sequence data can be an interesting future work.
> + Our Primal-Attention uses feature maps in primal, avoiding kernel computations in dual. However, it is not always easy to obtain or approximate feature maps of the kernel function, e.g., Gaussian kernel[3]. We applied feature maps of Cosine similarity kernels (see Remark 4.1 in paper). Although we achieved good performances in many datasets, more variants of feature maps can benefit wider applications.
>
> Thanks for this suggestion. We will elaborate it in the possible final version.
>
> [1] Allen-Zhu, Z., and Li, Y. "Backward feature correction: How deep learning performs deep learning." arXiv, 2020.
>
> [2] Tonin, F., Tao, Q., Patrinos, P., and Suykens, J.A.K. "Deep Kernel Principal Component Analysis for Multi-level Feature Learning." arXiv, 2023.
>
> [3] Rahimi, A., and Recht, B. "Random features for large-scale kernel machines." NeurIPS, 2007.

---

> > ### Comment · Reviewer_8hph · 2023-08-18
> > **More Questions on the Efficiency Advantage of Primal Attention**
> >
> > Thank you for your response. Could you please give further clarifications for the following point?
> >
> > 1. The memory advantage of Primal + DeiT-Small over the baseline DeiT-Small is not very significant while both obtain similar accuracies. Given this result, it is hard to claim that Primal + DeiT-Small is more efficient than the baseline. Can you also provide the running time analysis of Primal + DeiT-Small v.s. the baseline DeiT-Small for this experiment on ImageNet-1K? Similarly, how about the memory usage and running time for models trained on WikiText-103?
> >
> > 2. Why does Primal Attention gain more advantage in memory usage and running time on UEA Time Series than on D4RL and ImageNet-1K tasks?

---

> > > ### Author Response · Authors · 2023-08-20
> > > **Response to further questions on efficiency**
> > >
> > > We thank the reviewer for the in-depth comments on further clarifications of the efficiency analysis. Below, we address the raised two points in detail.
> > >
> > > ### R.1 Efficiency analysis on large-scale datasets
> > > **Experiment setups and results**
> > >
> > > For ImageNet-1K and WikiText-103, we provide the training memory and time on a single V100 GPU in Table 1.1 and Table 1.2 below. In the experiment, we adopt the architecture of "Primal.$+$Backbone", where the last attention layer of the Transformer is replaced by our Primal-Attention. As in large-scale data and complicated tasks, less information compression can be desired in the learning, especially in shallow layers, we thereby implement our KSVD-based Primal-Attention in the deep layer. Relevant explanations can refer to the 1st paragraph in Sec.5 in paper.
> > > + *ImageNet-1K:* While attaining the same accuracy and an enhanced low-rank property (Fig.1 in the paper), the efficiency gain of our Primal.$+$DeiT-Small/16 is limited, as we only replace the last layer of the 12-layer baseline with our Primal-Attention. Hence, it is reasonable that the efficiency improvement is less significant than that of UEA and LRA where we either implement both layers in the 2-layer baseline by Primal-Attention ("Primal."), or replace the 2nd layer ("Primal.$+$").
> > > + *WikiText-103:* Our "Primal.$+$Trans." shows similar efficiency as the 6-layer baseline, but significantly reduces the perplexity by $2.0$. Compared to the fine-tuned SoTA Flowformer (see Table 2 in our rebuttal), our "Primal.$+$Trans" is slightly inferior in perplexity by a small margin of 0.2, however, Flowformer is significantly less efficient by requiring $28.8 $% more running time than ours. The current "Primal.$+$Trans." is simply with default setups, better perplexity can still be expected by further tuning.
> > >
> > > **Table 1.1:** Efficiency analysis on ImageNet-1K.
> > > |ImageNet-1K|Top-1|Memory(GB)|Time(s/1K-steps)|
> > > |:-:|:-:|:-:|:-:|
> > > |DeiT-Small/16|79.8|14.2|2425.5|
> > > |Primal.$+$DeiT-Small/16|79.8|14.0|2330.2|
> > >
> > > **Table 1.2:** Efficiency analysis on WikiText-103.
> > > |WikiText-103|Perplexity($\downarrow$)|Memory(GB)|Time(s/1K-steps)|
> > > |:-:|:-:|:-:|:-:|
> > > |Trans.(2017)|33.0|9.0|3108.4|
> > > |Flow.(2022)|30.8|10.5|3998.4|
> > > |Primal.$+$Trans.|31.0|8.9|3104.0|
> > >
> > > **Further remarks**
> > >
> > > In summary, the efficiency gain of Primal-Attention implemented networks over baselines is influenced by two main factors: *1)* number of Primal-Attention layers used in the architecture, the more the better; *2)* sequence length of the training data, the longer the more significant.
> > > + With deep architectures, the efficiency can be further improved by replacing more layers with our Primal-Attention. Yet, in very deep Transformers, Primal-Attention is not necessarily always superior in performance when being applied to all layers, as the learning in shallow layers may not enjoy the benefits from the low-rank property from KSVD as much as the higher layers do. It would be interesting to explore a more generic implementation setup for Primal-Attention in very deep Transformers, as briefly mentioned in the last paragraph of possible future work in the Supplementary Material.
> > > + The length of the data sequence, i.e., $N$, is also a key influencing the efficiency. By avoiding the computation of the $N\times N$ attention matrix, our Primal-Attention can gain better efficiency  on longer-sequence datasets. Although ImageNet-1K is large-scale, currently Transformers treat each image as a sequence of length 197 (with $\tt cls$ token), which is actually not too long (even compared to some UEA datasets as shown in Table 3 in our rebuttal). Hence, this is also a reason why our Primal.$+$DeiT-Small/16 does not improve the efficiency significantly. Similarly in WikiText-103, the data sequence length is 512, which is also not really long, hence the efficiency of our "Primal.$+$Trans." is not always superior under the current setups.
> > >
> > > ### R.2 Efficiency gain of "Primal.$+$" in different tasks
> > > The efficiency gain of "Primal.$+$" over baseline is more significant on UEA and LRA, as the backbone has only 2 layers, hence replacing one layer makes a difference to the overall architecture. Moreover, UEA and LRA in general have longer training sequence length, which would signalize Primal-Attention's efficiency. In contrast, the backbones on D4RL, WikiText-103 and ImageNet-1K have more layers where canonical self-attention layers are the majority structures in "Primal.$+$" as shown in Table 2.1 below. Besides, the efficiency gain is less significant also due to the shorter training sequence lengths on these datasets. Explanations can refer to the response R.1 above.
> > >
> > > **Table 2.1:** Architecture of Primal.$+$.
> > > |Primal.$+$|canonical_layer+[primal_layer]|num_head|head_dim|
> > > |:-:|:-:|:-:|:-:|
> > > |UEA|1+[1]|8|64|
> > > |LRA|1+[1]|2|32|
> > > |D4RL|2+[1]|4|64|
> > > |WikiText-103|5+[1]|8|64|
> > > |ImageNet-1K|11+[1]|6|64|

---

> > > > ### Comment · Reviewer_8hph · 2023-08-21
> > > >
> > > > Many thanks for the prompt response. The authors have addressed my concerns. I will retain my score.

---

### Official Review · Reviewer_2pAx · 2023-07-07

**Soundness:** 2 fair
**Presentation:** 2 fair
**Contribution:** 2 fair
**Rating:** 4
**Confidence:** 4

**Summary:**

This paper proposes a Primal-Attention method to realize self-attention blocks in Transformer with a kernel matrix. It firstly explains the relationships between self-attention and asymmetric kernel matrix. Secondly it formulates self-attention in the form of kernel SVD and derive its primal and dual representations. And eventually it proposes Primal-Attention, which uses not only the projection scores involving the right singular vectors of the asymmetric attention kernel K, but also the ones involving the left singular vectors of the asymmetric attention kernel K. Besides, it conducts several experiments on several tasks.

**Strengths:**

1. The motivation to formulate a neural network structure into a kernel one is quite interesting and meaningful.
2. The primal-dual representations are derivable.


**Weaknesses:**

1. The explanation of why the projection scores involving the left singular vectors of the asymmetric attention kernel K are used is not adequate. It lacks enough analyses to prove their importance.
2. The results of experiments show little improvements compared with baseline methods. And only LRA benchmark has efficiency analyses.


**Questions:**

1. Why are the projection scores involving the left singular vectors of the asymmetric attention kernel K so important? What does it mean physically by doing so?
2. Why do you conduct experiments of efficiency only on LRA benchmark? And why does Primal+ have less efficiency than Primal+Trans?


**Limitations:**

Yes

---

> ### Author Rebuttal · Authors · 2023-08-08
>
> We thank the reviewer for the valuable comments. Below, we address the raised two main concerns in detail.
>
> ***R.1 Incorporating projection scores involving left singular vectors of the asymmetric attention kernel matrix $K$.***
>
> We provide detailed explanations and empirical evidences below. Relevant context can refer to line 164-182 on page 5 in paper.
> + In Section 2 in paper, we present the self-attention weights $K_{ij}=\kappa(x_i, x_j)=\text{softmax}(\left< q( x_i),k(x_j) \right>/\sqrt{d_k})$, where $K$ can be regarded as a kernel in relation to queries and keys via asymmetric measures $\kappa(x_i, x_j)\neq\kappa(x_j,x_i)$.  Given an asymmetric matrix $K$, there naturally exists information from two ways w.r.t. row space and column space [1]. This is in contrast to symmetric cases, e.g., Kernel PCA that only explores row data with a symmetric kernel [2]. Moreover, from the low-rank approximation perspective of KSVD to the self-attention kernel, utilizing both left and right singular vectors leads to the optimal approximation [3]. Hence, incorporating both sides of singular vectors could provide more possibilities to exploit comprehensive information, so as to benefit the learning of asymmetric kernels for boosted performances.
> + Empirically, we conduct an ablation study with (w/) and without (w/o) projection scores, i.e., $r$-scores, involving the left singular vectors.  Table 2pAx-1 shows that using both projections (w/ $r$-scores) helps boost performances, verifying our effectiveness in learning with asymmetric self-attention kernels.
>
> **Table 2pAx-1:** Ablation on the projection scores, i.e.,$r$-scores, involving left singular vectors on LRA with Top-1 test accuracy (%).
> |Primal.|ListOps|Text|Retrieval|Image|Pathfinder|Avg.Acc.|
> |:-:|:-:|:--:|:-:|:-:|:-:|:-:|
> |w/o $r$-scores|36.8|52.4|58.2|30.5|50.2|45.6|
> |w/ $r$-scores|37.3|61.2|77.8|43.0|68.3|**57.5**|
>
> |Primal.$+$Trans.|ListOps|Text|Retrieval|Image|Pathfinder|Avg.Acc.|
> |:-:|:-:|:--:|:-:|:-:|:-:|:-:|
> |w/o $r$-scores|37.1|65.1|79.2|42.8|72.8|59.4|
> |w/ $r$-scores|37.3|65.4|81.0|43.9|74.3|**60.4**|
>
> ***R.2.1 Performance improvement***
>
> In Tables 1, 2, 4 and 5  in paper, our method beats all compared methods including the very recent Transformers. In Table 1 on UEA Time Series, our method significantly improves over the baseline Transformer and also better than most other Transformers with a clear margin. Although our improvement is less distinctive than the very latest Flowformer (Flow.), it is reasonable to have comparable results on this simple benchmark with small-scale and short-sequence data. On LRA, D4RL and ImageNet-100 in Tables 2, 4 and 5 with more complex datasets, ours shows more substantial improvements. Notably, in reinforcement learning, our method has distinctively better performance than Flow. by a large margin (4.0%) in Table 4.
>
> ***R.2.2 Efficiency analyses***
>  + Long-range Arena (LRA) is a popular benchmark for evaluating efficiency in Transformers due to its long data sequence, which is a key element determining the kernel size and computation efficiency. Hence, we test on LRA in paper.
>  + In Table 3, "Primal." indicates that our Primal-Attention is applied to all attention layers, while in "Primal.$+$Trans." only the last layer in Transformers is replaced by Primal-Attention, as explained in line 219-229 on page 6. Thus, "Primal." shows to be more efficient than "Primal.$+$Trans.".
>  + We also present efficiency analyses on other datasets with comparisons to the baseline Transformer, and the most recent state-of-the-art Flowformer.
>
>  **UEA Times Series**: Our methods show the best efficiency among all compared methods. “Primal.$+$Trans.” improves over baseline Transformers (Trans.) by decreasing peak memory from 10.9GB to 6.5GB, while our “Primal.” even reduces it to 2.7GB and becomes more efficient with longer sequences, e.g., EthanolConcentration and SelfRegulationSCP2.
>
>  **D4RL Reinforcement Learning**: Our "Primal.$+$DT" achieves comparable time and memory efficiency as the baseline Decision Transformer (DT), while Flowformer (Flow.) shows significantly lower efficiency. Recall that in Table 4  in paper, our method achieves a much better average reward of 77.5 than DT (72.2) and Flow. (73.5).
>
> **Table 2pAx-2:** Comparisons on running time (s/Epoch) and memory consumption (GB) on UEA Time Series.
> |UEA benchmark (sequence length)|Trans.|Flow.|Primal.+Trans.|Primal.|
> |:-:|:-:|:-:|:-:|:-:|
> |EthanolConcentration (1751)|4.3|2.4|3.3|2.3|
> |FaceDetection (62)|10.7|9.8|5.7|6.9|
> |HandWriting (152)|0.3|0.3|0.3|0.4|
> |HeartBeat (405)|0.7|0.7|0.7|0.4|
> |JapaneseVowels (26)|0.5|0.6|0.5|0.5|
> |PEMS-SF (144)|0.6|0.7|0.7|0.8|
> |SelfRegulationSCP1 (896)|1.8|1.4|1.6|1.4|
> |SelfRegulationSCP2 (1152)|1.8|1.3|1.6|1.3|
> |SpokenArabicDigits (83)|3.7|4.4|4.0|4.5|
> |UWaveGestureLibrary (315)|0.3|0.3|0.3|0.4|
> |**Avg. Time (s/Epoch)**|2.5|2.2|**1.9**|**1.9**|
>
> |UEA benchmark|Trans.|Flow.|Primal.+Trans.|Primal.|
> |:-:|:-:|:-:|:-:|:-:|
> |**Memory (GB)**|10.9|2.8|6.5|**2.7**|
>
> **Table 2pAx-3:** Comparisons on running time (s/1K-steps) and memory consumption (GB) on D4RL.
> |Time |Medium-Expert|Medium|Medium-Replay|
> |:-:|:-:|:-:|:-:|
> |DT (reward: 72.2)|20.8|20.8|20.8|
> |Flow. (reward: 73.5)|54.4|54.4|54.3|
> |Primal.+DT (**reward:77.5**)|23.5|23.4|23.3|
>
> |Memory|Medium-Expert|Medium|Medium-Replay|
> |:-:|:-:|:-:|:-:|
> |DT|0.3|0.3|0.3|
> |Flow.|1.5|1.5|1.5|
> |Primal.+DT|0.3|0.3|0.3|
>
> [1] Suykens, J.A.K. "SVD revisited: A new variational principle, compatible feature maps and nonlinear extensions." Applied and Computational Harmonic Analysis, 2016.
>
> [2] Schölkopf, B., Alexander S., and Klaus-Robert M. "Nonlinear component analysis as a kernel eigenvalue problem." Neural computation, 1998.
>
> [3] Eckart, C., and Gale Y. "The approximation of one matrix by another of lower rank." Psychometrika, 1936.

---

> > ### Comment · Reviewer_2pAx · 2023-08-19
> >
> > Thanks for your response. I still have some questions.
> > In the Table 2pAx-1, the performance of Primal with both projections is significantly worse than the Primal+Trans without both projections but has projection scores involving right singular vectors in Transformer, is it means that the projection scores involving the left singular vectors has bad influence on the performance?
> > On the other hand, from the results of Primal+Trans in Table 2pAx-1, utilizing the both projections only has slightly better performance on two tasks, while shows negligible improvements on the remaining three task. The necessity of utilizing left singular vectors should be explained more intuitively.

---

> > > ### Author Response · Authors · 2023-08-20
> > > **Further response to the benefits of left singular vectors**
> > >
> > > We thank the reviewer for the reply. We provide the following 3 aspects to address your concerns and eliminate possible confusions.
> > >
> > > ### 1. Projection scores involving left singular vectors are beneficial to the performance.
> > >
> > > For clarity, in Table R.1 we summarize the setups and architectures of Table 2pAx-1. We use the 2-layer backbone baseline commonly used on LRA. To make the ablation more comprehensive, we **respectively** conduct this study on the two main architectures in this work, i.e., Primal. and Primal.$+$Trans. (see the 1st paragraph of Sec.5 in the paper for details).
> > >
> > > **Table R.1:** Ablation setups on LRA.
> > > |Primal.|w/o $r$-scores|w/ $r$-scores|
> > > |:-:|:-:|:-:|
> > > |Layer 1|[right]|[right;left]|
> > > |Layer 2|[right]|[right;left]|
> > > |Avg. Acc.|45.6|**57.5**|
> > >
> > > |Primal.$+$Trans.|w/o $r$-scores|w/ $r$-scores|
> > > |:-:|:-:|:-:|
> > > |Layer 1|canonical attention|canonical attention|
> > > |Layer 2|[right]|[right;left]|
> > > |Avg. Acc.|59.4|**60.4**|
> > >
> > > The two subtables in Table 2pAx-1 are the ablation study of using the left singular vectors on **two different network architectures respectively**, so that their results are not directly comparable under the setups in this ablation. Therefore, it is not the left singular vectors that bring bad influence on the performance, and such evaluations should be considered separately in these two subtables. More specifically,
> > >
> > > 1. Using the projections of left singular vectors, i.e., (w/ $r$-scores), is helpful to the performance, when Primal-Attention is applied to all layers (Primal.) and also when it is applied only to deep layer (Primal.$+$Trans.). The role of the left singular vectors is more influential in Primal. for boosting the performance, where all canonical attention is replaced with the KSVD-based Primal-Attention.
> > > 2. As mentioned by the reviewer, Primal.(w/ $r$-scores) has inferior performance than Primal.$+$Trans.(w/o $r$-scores). This is due to: Primal.(w/ $r$-scores) applies low-rank KSVD to the first layer, where the learning in this shallow layer does not always desire for the low-rank property from KSVD as much as the deep layer does on LRA.
> > >
> > > ### 2. Performance gain is substantial.
> > >
> > > From the results of Primal.$+$Trans. in Table 2pAx-1, the mentioned performance improvement is  substantial on LRA.
> > > 1. *We obtain 1% gain (59.4% $\to$ 60.4%) in average accuracy:* In Table 2 in paper, recent SoTA method YOSO-E has only 0.4\% gain compared to the baseline. Hence, 1% gain can be regarded as substantial on LRA.
> > > 2. *LRA benchmark should be evaluated as a whole:* Since there are five different tasks in this benchmark, the average accuracy is the most conclusive measure, which is also attached great importance by almost all SoTA methods. In Table 2 in paper, YOSO-E even has inferior performance on Pathfinder dataset than baseline. However, this cannot diminish its overall better performance on this benchmark.
> > > 3. *In each individual task, using both projection scores consistently brings improvement*: As in Table R.2, the accuracy gain is significant on at least 3 tasks: Retrieval, Image and Pathfinder; besides, using both projection scores consistently improves performances in all tasks, which is substaintial for the LRA benchmark.
> > >
> > > **Table R.2:** Acc.(%) gain of Primal.$+$Trans.(w/ $r$-scores) over Primal.$+$Trans.(w/o $r$-scores).
> > > |Primal.$+$Trans.|ListOps|Text|Retrieval|Image|Pathfinder|Avg.Acc.|
> > > |:-:|:-:|:--:|:-:|:-:|:-:|:-:|
> > > |Acc.$\uparrow$|0.2|0.3|1.8|1.1|1.5|1.0|
> > >
> > > ### 3. Necessity remarks
> > >
> > > 1. **Theoretically,** the derived KSVD can be regarded as a low-rank approximation to the self-attention kernel matrix where utilizing both left and right singular vectors leads to the optimal approximation.
> > > 2. **Experimentally,** using both projection scores brings substantial performance gain than using only one.
> > > 3. **Intuitively,** using both projection scores can be treated as considering both directions in a directed graph. Section 2 in the paper presents the asymmetric attention weight $K_{ij}=\kappa(\boldsymbol x_i, \boldsymbol x_j)=\text{softmax}(a(\boldsymbol x_i, \boldsymbol x_j))$ with $a(\boldsymbol x_i, \boldsymbol x_j)=\left<q(\boldsymbol x_i), k(\boldsymbol x_j)\right>/\sqrt{d_k}$. This shows that the attention kernel $K$ can be interpreted as message passing in a directed graph in relation to queries and keys, with asymmetric similarity measures $\kappa(\boldsymbol x_i, \boldsymbol x_j)\neq\kappa(\boldsymbol x_j, \boldsymbol x_i)$. Using only the right singular vectors means considering only one directionality, where additional information might reside in another directionality to enhance the learning on such an asymmetric kernel matrix. Useful references can be checked on page 128-129 in [1] for graph theory. In the asymmetric case, it is natural to consider directed information, i.e., both right and left singular vectors as in our case, for enhancing performance.
> > >
> > > [1] Estrada, E. "The structure of complex networks: theory and applications." Oxford University Press, 2012.

---

### Author Rebuttal · Authors · 2023-08-09

Dear Program Chairs, Area Chairs, and Reviewers,

First of all, we would like to thank you for your time and valuable comments, which help improving our work.

In this work, we provide a new framework to interpret self-attention in Transformers via asymmetric Kernel Singular Value Decomposition (KSVD). Our work addresses the intrinsic asymmetry residing in self-attention and fills the gap of most of existing works on Transformers resorting to the classic techniques using symmetric Mercer kernels. With KSVD, a primal-dual representation of self-attention is formulated and its corresponding primal and dual optimization problems are also cast by maximizing the projection variances in the attention outputs. Based on the derived analytical results, a new attention mechanism is proposed with improved efficiency and performances, namely the Primal-Attention, and its optimization is also incorporated through a regularization term with enhanced low-rank properties.

We are grateful that the reviewers unanimously regard our work as novel and interesting towards understanding self-attention and of potentially high-impact. The review comments mainly raise to present more empirical evaluations and to elaborate some technical details. In the rebuttal, we have addressed the comments from all reviewers point-wisely. To be more specific, *i)* we present more experimental results that all support the superiority of our method, including efficiency analyses, large-scale datasets, ablation on the incorporation of projection score involving the left singular vectors (to **Reviewer 2pAx, Reviewer 8hph**); *ii)* we provide further explanations to better introduce the primal and dual optimization in Eq.(6)-(7) and the objective of Primal-Attention in Eq.(10)-(11) based on the property from stationary conditions in Lemma 4.2 (to **Reviewer aphr**).

We hope that our detailed responses could well address the comments and be assessed by Chairs and Reviewers. We sincerely look forward to further discussions with the reviewers.

Best wishes,

Anonymous author(s) of Paper3761

---

### Decision · Program_Chairs · 2023-09-21

**Decision:**

Accept (poster)

**Comment:**

Growing body of work aims to improve self-attention by treating it as a kernel machine. However, they utilize symmetric kernels. While the symmetry issue might be perceived as a minor concern, canonical self-attention is inherently asymmetric through distinct key-query embeddings. The main contribution of this paper is representing self-attention through asymmetric Kernel Singular Value Decomposition (KSVD). Building on this, the authors provide a primal-dual representation of self-attention and introduce the so-called Primal-Attention through the primal representation of KSVD.

The reviewers and I found the paper to be a solid contribution with strong in its theoretical underpinnings. The authors also demonstrate the empirical value of the proposed Primal-Attention. There are some notable shortcomings, primarily related to insufficient explanations and empirical analyses. The reviewers collectively pointed to the need for greater clarity in outlining methodologies and for more comprehensive experimental evaluations. The authors have addressed many of these shortcomings during the response period, for instance, they have provided more thorough experiments. Overall, despite some concerns, the contributions and theoretical strengths of the paper warrant an "Accept" decision. I recommend reviewers to address any remaining weaknesses to enhance the paper's impact and rigor. Specifically, the authors should

* provide clear explanations for the critical methodologies and concepts that reviewers found lacking.
* carefully incorporate new empirical findings and
* address the omitted citation and any other relevant literature that strengthens the theoretical basis of the paper.